# TabAgent: A Framework for Replacing Agentic Generative Components with Tabular-Textual Classifiers

## Abstract

Agentic systems, AI architectures that autonomously execute multi-step workflows to achieve complex goals, are often built with LLM-based control flows where agents make repeated decisions such as routing, shortlisting, gating, and verification, each drawn from a closed set of options. This approach simplifies development but makes deployments slow and expensive as these decision components are invoked repeatedly and their latency/tokens accumulate. We propose **TabAgent**, a general framework for replacing generative decision components in closed-set selection tasks with a compact textual-tabular classifier trained on signals extracted from execution traces, by (i) automatically distills schema/state/dependency features from trajectories **TabSchema**, (ii) augments coverage with schema-aligned synthetic supervision **TabSynth**, and (iii) scores candidates with a compact textual tabular classifier **TabHead**. This approach enables agentic systems to automatically optimize their own decision-making by learning from successful executions, allowing for seamless replacement of expensive generative components. On long-horizon, cross-application benchmark (AppWorld), TabAgent maintains task-level success while eliminating shortlist-time LLM calls and reducing latency (by $\sim 95\%$) and inference cost (by 85–91%). Beyond tools shortlisting, TabAgent extends to application selection, establishing a paradigm for learned discriminative replacements of generative bottlenecks in production agentic architectures. Upon acceptance, we will release the full code.

## 1 Introduction

Agentic systems promise to turn natural-language goals into end-to-end action across applications and modalities, but deployments contend with two coupled pressures: latency from long-context reasoning and cost from repeated long-context LLM calls, a consequence of the prevalent practice of modeling each system component as a frontier LLM call (Schneider, 2025). The pressure is sharpest at decision points on the execution path, most notably *tool shortlisting*, where a "tool" is any externally callable capability (API endpoint, function, or GUI operation) with a typed schema and observable outputs/side effects. The agent must select a small subset from tool catalog that typically span hundreds to thousands of tools (Li et al., 2023; Wang et al., 2024a; Qu et al., 2025), where the shortlisting task can incur a substantial cost and take more than one minute to complete in multi-step long-horizon tasks Gonzalez-Pumariega et al. (2025); Aghzal et al. (2025).

Although efforts to improve tool shortlisting efficiency have been made, the fundamental tension between selection accuracy and computational cost remains unresolved. Traditional accelerations prune the search space with fast retrieval and hierarchy-aware heuristics, which improve performance but remain state-/set-/structure-blind when preconditions, session memory and phase, argument schemas, and inter-tool dependencies matter (Sparck Jones, 1972; Robertson et al., 2009; Karpukhin et al., 2020a; Anantha et al., 2023; Zheng et al., 2024). Recent optimizations use smaller LLMs or pair two-stage retrieval with LLM refinement, reducing per-call cost yet preserving the multi-call decision loop that drives latency and cost as plan depth and tool catalog size grow (Qin et al., 2023). Alternative lines as learning-to-rank, graph-based navigation, or embedding-only classifiers, often optimize offline proxies (e.g., $\text{Recall}@K$), under task state (Zheng et al., 2024; Liu

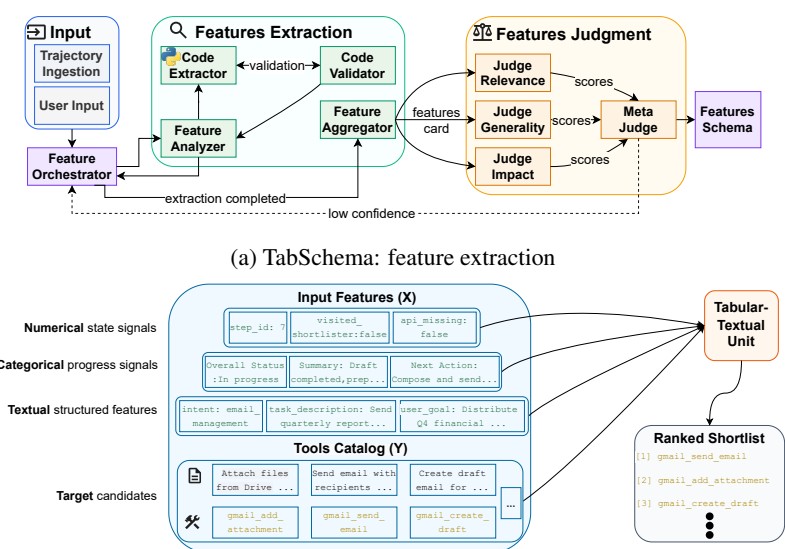

(a) TabSchema: feature extraction

(b) TabHead: single-pass classifier

Figure 1: (a) TabSchema compiles trajectory-derived schema, state, and dependency signals into tabular features. (b) TabHead consumes features and candidates to output calibrated probabilities.

et al., 2024; Moon et al., 2024). However, these approaches either continue with the generative paradigm or omit important trajectory signals, and show limited generalization to unseen tools.

TABAGENT addresses this need by reframing tool shortlisting (and other decision heads such as task-complexity and application-identifier prediction) as tabular classification and replacing the GenAI shortlister component with a single compact decision module. The framework comprises two agentic sub-architectures: **TabSchema**, a hierarchical behavior abstraction pipeline, compiles execution traces into a tabular schema and derives structured features for each (context, tool) pair that capture schema (taxonomy depth, argument types, I/O cardinality), state (thought traces, plan-precondition satisfaction, resource availability), and dependencies (co-usage, precedence, success/failure signals). See Fig. 1a. **TabSynth** then generates schema-aligned synthetic examples to expand coverage and supervise the classifier for the task. Finally, **TabHead**, a compact tabular+text classifier, consumes the TabSchema features together with the text of each candidate option and outputs a calibrated probability $p(o \mid \text{context})$ for $o \in \mathcal{C}$ (Fig. 1b). Our approach builds on the hypothesis that agentic systems inherently generate rich behavioral features through self-supervised interaction with environments and tools, creating execution traces that encode decision-relevant signals in a structured, tabular-representable form. Crucially, this enables a practical deployment pattern where systems can continuously improve their own efficiency by: accumulating successful traces, extracting behavioral signals, training compact classifier for generative bottlenecks, and performing hot-swaps during operation without manual intervention At inference, a single $\sim$50M-parameter TabHead forward pass produces the decision, yielding reduced latency and cost while maintaining task-level performance.

Our experiments evaluate TABAGENT as a drop-in replacement for the GPT-based shortlister within IBM CUGA (Marreed et al., 2025), the SOTA open source agent on AppWorld. Replacing only the shortlist head preserves shortlist quality: across all five applications we obtain Recall@7 $\geq 0.88$ and Recall@9 $\geq 0.92$, and adding schema-aligned synthetic supervision lifts macro $P@R$ by +0.14 on average. Relative to CUGA's GPT-4.1 shortlister, TABAGENT eliminates shortlist-time LLM calls and markedly improves efficiency: a single 50M-parameter pass executes in $2.682$ ms at $\$2.0 \times 10^{-7}$ per read versus $7.50$ s and $\$0.052$ for GPT-4.1, i.e., at least a 95% latency reduction and at least an 85% cost reduction in our setup (Fig. 2; App. Table 12). Experiments indicate that TabSchema features account for most performance gains over using the task description alone, and that schema-aligned synthetic supervision expands coverage of low-frequency schema and dependency patterns, improving macro $P@R$ by +0.14 on average. These findings support the hypothesis that behavior-grounded signals extracted from execution traces can replace closed-set generative decision heads with a single compact tabular classifier. Our work makes three key contributions:

1. **TabAgent** First framework to abstract agentic decision heads into a tabular representation and systematically replace generative modules with a compact classifier.

2. **Automatic schema engineering** Extraction of schema, state, and dependency signals from trajectories and tool I/O, together with schema-aligned synthetic supervision that removes manual labels and enables rapid adaptation.

3. **Production-scale validation** In IBM CUGA (a state-of-the-art open-source system), replacing only the tool shortlisting generative component preserves shortlist quality while removing shortlist-time LLM calls and yielding substantial latency and cost reductions.

## 2  RELATED WORK

The prevailing benchmarks evaluate tool selection largely in isolation: Qin et al. (2023) curate TOOLBENCH ($\sim$ 16k APIs), Li et al. (2023) release API-BANK with multi-turn dialogues, and Song et al. (2023) propose RESTBENCH with human-annotated API call chains. Tang et al. (2023) provide TOOLALPACA for instruction tuning, and Du et al. (2024) stress hierarchical retrieval at large catalog scale. These resources offer standardized correctness metrics (e.g., Recall@K, pass/win) Schütze et al. (2008), however they emphasize offline retrieval or scripted execution and rarely measure shortlist time efficiency, wall-clock latency under SLOs, token spend, or test cross-application dependencies and multi-agent coordination. We therefore evaluate on long-horizon, cross-application environments Trivedi et al. (2024), where the shortlisting task lies on the critical path toward achieving the goal, and its efficiency directly affects end-task success, latency and cost.

Early approaches treat tools as documents and apply sparse or dense retrieval: TF–IDF/BM25 (Sparck Jones, 1972; Robertson et al., 2009) and Sentence-BERT/DPR/ANCE (Reimers & Gurevych, 2019; Karpukhin et al., 2020a; Xœiong et al., 2020). These provide semantic matching but are state-blind (ignore task progress and preconditions), set-blind (score items independently rather than complementary sets), and structure-blind (neglect hierarchies and argument schemas), limiting precision in long-horizon, multi-step, dependency-heavy tasks. Building on this, modern retrieval-style methods inject structure and supervision. Anantha et al. (2023) learn prototypical representations and progressive matching, reporting increment in accuracy over a ChatGPT baseline. Zheng et al. (2024) add hierarchy-aware reranking with seen/unseen adaptive truncation to improve execution fidelity. Qu et al. (2024) optimize completeness, ensuring all tools required to solve the task appear in the shortlist. These methods improve shortlist quality but add multi-stage inference whose latency scales with tools catalog size. In parallel, Moon et al. (2024) cast shortlisting as supervised classification over (query, tool) embeddings, however, this design omits explicit schema/state/dependency signals and evaluates primarily with static retrieval, offering limited evidence for interactive, long-horizon tasks or shortlist-time efficiency.

LLM-centric approaches train or prompt the agent to choose tools directly. Qin et al. (2023) fine-tune 7–13B-class controllers and traverse tool abstraction levels (Category $\rightarrow$Tool$\rightarrow$API) with DFS, improving tool-use success but typically incurring multiple LLM calls per task. Liu et al. (2024) organize massive toolsets as a directed graph and use an LLM to navigate successors while an evaluator updates transition weights, scaling to large libraries yet compounding calls with depth. Gao et al. (2024) apply curriculum learning over tool complexity, however generalization to unseen tools remains fragile and inference relies on LLMs. Overall, these approaches achieve accuracy at substantial computational cost, via chain/tree search and repeated decoding, while overlooking multi-agent coordination and long-horizon task signals that are essential for robust tools selection.

Classical compute reduction techniques, such as distillation, pruning, quantization, parameter-efficient tuning, lower per-call compute and using small LMs leave unchanged the number of generations required by search-based shortlisters (Hinton et al., 2015; Sanh et al., 2020; Hu et al., 2021; Belcak et al., 2025) . We instead reduce compute at the architectural level by replacing generative components in agentic systems with a compact tabular classifier Galke et al. (2022); Saattrup Nielsen et al. (2025). Tabular architectures that ingest text features directly demonstrate the effectiveness of specialized inductive biases for structured data (Huang et al., 2020; Hollmann et al., 2025) . Concretely, we adapt Arazi et al. (2025), a 50M-parameter tabular model that extends this tabular domain with target-aware tokens and a high-quality training corpus, achieving SOTA performance.

## 3 TABAGENT

TABAGENT addresses closed-set decision heads in agentic systems, components that repeatedly select from finite option sets such as tools, applications, routers, or strategies, by learning discriminative boundaries from successful logged execution traces. Our approach rests on the hypothesis that execution traces from successful agentic systems encode decision-relevant signals in a structured, tabular-representable form that compact discriminative classifiers can extract and reproduce. Concretely, if an LLM successfully solved a closed-set decision by processing agentic context, that same context (now tabularized) can teach a compact discriminative classifier to reproduce the decision boundary at a fraction of the cost and latency Hsieh et al. (2023). By construction, the pipeline is agent-agnostic as it binds only to execution traces and typed schemas, not to any particular agent architecture, model, or prompt, enabling the same reduction (trace $\rightarrow$ features, candidates $\rightarrow$ labels) to apply across decision heads by swapping the label space and feature card (App. §I).

### 3.1 TABSCHEMA: AGENTIC FEATURE EXTRACTION ARCHITECTURE

TabSchema employs a hierarchical agentic orchestration to extract behavior-grounded features from agent trajectories, ensuring both comprehensive coverage and explainable feature selection through multi-stage validation and judicial review (Zhang et al., 2025). The **FeatureOrchestrator** coordinates agents and maintains a ledger while routing a trajectory batch $\mathcal{T}$ into the **FeatureAnalyzer**, which first designs feature specifications before any code is written. Concretely, the analyzer enumerates (i) extractable signals lifted directly from traces (e.g., thought snippets from a specific node, last return status, error types), and (ii) computed signals derived from trace structure (e.g., step index within the agentic loop, "has agent $X$ been visited?", co-usage windows), attaching for each candidate $f$ its type, units/range, provenance, and a task-specific relevance score with a concise relevance description (Ahn et al., 2022). We adopt LLM-based feature extraction since the trajectory reflects another LLM's decision-making process, making LLM-driven explainability an established practice for surfacing decision-relevant signals (Kroeger et al., 2023). The specification then flows to a **CodeExecutor**, which implements feature code in a sandboxed runtime. Upon compilation/runtime failure, the executor performs up to three retries that concatenate the prior error traces and fixes to inform the next attempt (Madaan et al., 2023; Lv et al., 2024). The resulting artifacts are inspected by a **CodeValidator** that authenticates feature validity and quality and returns a structured validation review to the executor when repairs are needed. Otherwise, the validator signals agreement that the extracted features follow the analyzer's plan. This validation emerged from observing unreachable and hallucinated features in early iterations; by verifying feature feasibility through executable code, TabSchema produces universal extractors that can be automatically deployed when integrating TabAgent into production systems (Renze & Guven, 2024). The analyzer finally observes realized features against its plan to decide between a refinement cycle or declaring completion.

Building on these cards, TabSchema performs a multi-LLM judicial review that acknowledges how quality dimensions (relevance, generality, impact) often diverge and how single judges may overlook important aspects (Zheng et al., 2023). In this process, three specialized judges independently read the features card and return a score in $[0, 1]$ with short justification. These judgments refine or confirm the analyzer's per-feature relevance score and description (Liu et al., 2023). The **MetaJudge** is realized as a structured prompt that reconciles the three rationales with analyzer statistics and issues a single, explained decision (accept/revise) (Li et al., 2025). This hierarchical orchestration evolved from managing iteration over hundreds of trajectories: separation of concerns (design $\rightarrow$ implement $\rightarrow$ validate $\rightarrow$ evaluate) enables iterative refinement, produces auditable feature cards, and catches errors early. This "LLM-as-a-judge" construction builds on evidence that LLM evaluators approximate human preferences (Li et al., 2024; Gu et al., 2024), and our extension applies this to feature selection rather than output evaluation. See prompts and output examples in App. D

### 3.2 TABSYNTH: SYNTHETIC DATA GENERATION

In practice, real trajectories under represent rare but decision critical patterns (e.g., argument-compatibility), which limits coverage when training a compact discriminative head. To close this gap, we introduce TabSynth, a schema-aligned augmentation module that expands supervision while preserving the behavior-grounded signals surfaced by TabSchema (diagram in Fig. 7). TabSynth consumes the feature cards produced by TabSchema and instantiates additional training rows for

each pair (context, candidate) constrained by three inputs: the FEATURES_CARD schema, TRAJECTORY_TABLE providing observed execution patterns, and TOOL_CATALOG_SUMMARY specifying taxonomy and argument as present in real data. Each synthetic row is a feature vector with populated features fields, validated by: (i) schema checks (types, arity, taxonomy level), (ii) dependency feasibility (no impossible chains), and (iii) de-duplication/diversity against real Wang et al. (2024b); Barr et al. (2025). We use a small budget (10 synthetic per candidate) and mix synthesized and real examples during training to avoid distributional drift. More details in App. G.

### 3.3 TABHEAD: TEXTUAL–TABULAR CLASSIFICATION

TabHead instantiates a recent SOTA textual–tabular classifier (Arazi et al., 2025) over two aligned views of each candidate. The tabular view is the TabSchema feature vector, and the candidate textual representation is a single string formed by concatenating the candidate name with its description and brief argument hints (Fig. 1b), see example in App. H. Training phase mixes real trajectories with schema-aligned TabSynth data to widen coverage while staying within the observed schema.

We handle three decision patterns through one pointwise reduction. For single-class tasks (e.g., task complexity identifier), we train a softmax over $\mathcal{Y}$. For multi-label problems (e.g., application selection), we use binary relevance: for each example $x$ and label $y \in \mathcal{Y}$, we build a row $(x, y)$ and predict $p(y \mid x)$; thresholding produces $\hat{Y}$ (Tsoumakas & Katakis, 2008). For ranking (e.g., tool shortlisting), we again score $(x, y)$ pairs: during training we include one row per correct candidate $y \in Y_t$, and at inference we sort all candidates by $p(y \mid x)$ and return the top-$K$ (Liu et al., 2009).

## 4 EXPERIMENTAL SETUP

### 4.1 SETUP SPECIFICATIONS

We evaluate TABAGENT on AppWorld (Trivedi et al., 2024), a long-horizon, cross-application benchmark with standardized multi-application evaluation. We focus on five applications, Amazon, Gmail, Phone, SimpleNote, and Spotify, selected based on task availability. These behave as complementary sub-benchmarks with heterogeneous action spaces and decision semantics, while also supporting cross-application composition. Each application exposes its own tool catalog with a distinct taxonomy and argument schema, differing in cardinality, type systems, and dependency structure (preconditions, precedence, co-usage). As a result, task distributions differ not only lexically but structurally (e.g., required-set size $R$, plan length, argument-type entropy, dependency density), and cross-app tasks require composing incompatible schemata under typed constraints. Detailed per-app statistics, divergence analyses, and cross-app composition appear in App. A.1.

Our goal is to replace an already validated and expensive generative component of agentic systems (e.g., tool shortlisting) with an efficient foundational discriminative head ($\sim 50M$ parameters), trained on features extracted from high quality trajectories, rather than to invent new component logic. This follows a practical TabAgent deployment lifecycle: i) deploys the agentic system with trajectory logging enabled, ii) collects execution data from successful tasks to establish baseline performance and extract TabSchema features, and iii) performs a hot-swap to the discriminative TabAgent head while preserving the surrounding agent architecture. To do so, we use IBM CUGA (Marreed et al., 2025), the SOTA open source agentic system on AppWorld, as the reference agent. We instrument CUGA and replace only its shortlisting module by extracting TabSchema features from its trajectories and training our tabular classifier TabHead on top of those features. As a text-only control, we also evaluate CODEACT as baseline (Lv et al., 2024) whose sole input is the task description, aligning with its natural-language–only interface and isolating the contribution of TabSchema features. Focusing on a single, strong agent keeps instrumentation consistent, removes confounding design choices across stacks, and aligns with our objective to replicate proven decision boundaries at lower cost and latency. Since AppWorld does not provide ground truth for tool shortlisting, we derive labels from successful CUGA trajectories. For each task $t$, the relevant set $G(t)$ is defined as the set of tools actually used in a canonical successful solution. This construction matches our aim of reproducing the choices that already lead to success. In total we work with 605 tasks for which CUGA yields successful trajectories, see App. B.

## 4.2 IMPLEMENTATION DETAILS

To train TabHead component we use the default parameters ($lr = 0.001$, $lora\_r = 16$, $max\_epochs = 50$). We compare TabAgent with three baseline families that represent common approaches to tool shortlisting. The first is sparse retrieval with Okapi BM25, a fast and widely used classical information retrieval model that scores tools by lexical overlap between the task description and tool documentation (Robertson et al., 2009; Sparck Jones, 1972).

The second family employs dense semantic retrieval (DSR) (semantic similarity) through a bi-encoder semantic setup (Karpukhin et al., 2020b) in three configurations: Zero-Shot ranks tools using off-the-shelf sentence embeddings without task-specific training, FT (Real) applies contrastive fine-tuning on real agent trajectories (Chen et al., 2020), and FT (Real+synth) combines real trajectories with TabSynth supervision during training. All variants use the same e5 encoder as TabHead for fair comparison (Wang et al., 2022). Fine-tuned variants optimize a contrastive objective with in-batch and mined hard negatives, learning to align task/context representations with relevant tool descriptions in a shared embedding space (Reimers & Gurevych, 2019). At inference, candidates are scored by inner-product between task and tool embeddings, returning the top-$K$ results.

The third family is generative LLM controllers. We evaluate Llama 3.2 1B/3B, and Llama 3.1 8B Dubey et al. (2024) as decision makers that consume CUGA's original shortlister state and the full tool catalog, then produce a ranked shortlist through instruction following. Each model processes the complete catalog for the active application, including tool names, argument signatures, and descriptions. Inference uses low temperature decoding with nucleus sampling for stability.

## 4.3 EVALUATION METRICS

We evaluate fine tuned models with 5 fold cross validation at the task level. Folds are stratified by application and by deciles of $|G(t)|$, the number of relevant tools per task. All feature rows and synthetic augmentations derived from a given task remain within a single fold to prevent leakage. In each round we train on 4 folds and evaluate on the held out fold, then macro average metrics across tasks over the 5 test folds. Non fine tuned baselines are evaluated without training. For methods with stochastic behavior, each task is run 5 times with distinct sampling seeds and task level scores are averaged. Unless noted otherwise, we report macro averages over tasks with 95% confidence intervals. Statistical significance is assessed by paired resampling over tasks with Holm Bonferroni control within each metric across the 5 applications. Further details appear in App. E.

TABAGENT instantiates multiple decision heads. Accordingly, we evaluate with metrics matched to head semantics. We consider: (i) single-class classification (e.g., task-complexity identification), (ii) multi-label classification (e.g., application identification), and (iii) top-$k$ for tool shortlisting. For Top-$k$ shortlisting, consider each task $t \in T$, let $G(t)$ denote the ground-truth relevant set of tools with size $R_t = |G(t)|$. Let $\pi(t)$ be the system's ranking over the catalog and $S_k(t)$ the top-$k$ prefix of $\pi(t)$. When scores tie, we apply a stable sort by tool id.

*Recall@$k$* evaluates recovery at a fixed shortlist budget:

$$\text{Recall@}k = \frac{1}{|T|} \sum_{t \in T} \frac{|S_k(t) \cap G(t)|}{|G(t)|}.$$

*R-precision (P@R)* adapts the cutoff to the task's relevant-set size and captures completeness:

$$P@R = \frac{1}{|T|} \sum_{t \in T} \frac{|S_{R_t}(t) \cap G(t)|}{R_t}.$$

$P@R$ equals Recall@$R$ because both reduce to $|S_{R_t}(t) \cap G(t)|/R_t$ per task. We use the $P@R$ name for IR consistency (Schütze et al., 2008). Throughout, we foreground $P@R$ for completeness claims and use Recall@$k$ to assess performance under fixed production budgets. See App.L.

## 5 RESULTS

Across five applications, **TabAgent (+synth)** leads or ties the strongest baseline, attaining Recall@7 $\geq 0.88$ in every domain and Recall@9 $\geq 0.92$ when narrowing catalogs of hundreds

| Method | R-precision (P@R) | | | | | Recall@7 | | | | | Recall@9 | | | | |
|---|---|---|---|---|---|---|---|---|---|---|---|---|---|---|---|
| | AZ | GM | PH | SN | SP | AZ | GM | PH | SN | SP | AZ | GM | PH | SN | SP |
| BM25 | 0.28 | 0.19 | 0.35 | 0.42 | 0.28 | 0.48 | 0.51 | 0.54 | 0.72 | 0.60 | 0.54 | 0.58 | 0.57 | 0.74 | 0.71 |
| DSR (Zero-Shot) | 0.39 | 0.18 | 0.30 | 0.55 | 0.30 | 0.65 | 0.46 | 0.82 | 0.89 | 0.62 | 0.67 | 0.53 | 0.88 | 0.96 | 0.70 |
| DSR-FT (Real-Only) | 0.45 | 0.52 | 0.57 | 0.63 | 0.46 | 0.89 | 0.84 | 0.95 | 0.99 | 0.82 | 0.90 | 0.87 | 0.95 | 0.99 | 0.85 |
| DSR-FT (Real+synth) | 0.69 | 0.56 | 0.85 | 0.92 | 0.64 | 0.93 | 0.90 | **0.98** | **1.00** | 0.85 | 0.95 | 0.93 | 0.99 | **1.00** | 0.88 |
| Llama-3.2-1B-Instruct | 0.02 | 0.03 | 0.06 | 0.07 | 0.09 | 0.29 | 0.27 | 0.29 | 0.45 | 0.33 | 0.32 | 0.29 | 0.33 | 0.46 | 0.34 |
| Llama-3.2-3B-Instruct | 0.38 | 0.34 | 0.40 | 0.42 | 0.39 | 0.60 | 0.58 | 0.63 | 0.76 | 0.64 | 0.66 | 0.66 | 0.66 | 0.79 | 0.67 |
| Llama-3.1-8B-Instruct | 0.56 | 0.45 | 0.38 | 0.70 | 0.67 | 0.56 | 0.50 | 0.65 | 0.71 | 0.67 | 0.56 | 0.50 | 0.65 | 0.73 | 0.69 |
| TabAgent (Real-Only) | 0.57 | 0.58 | 0.66 | 0.62 | **0.70** | 0.91 | 0.88 | **0.98** | **1.00** | **0.88** | 0.93 | 0.91 | **1.00** | **1.00** | 0.90 |
| TabAgent (Real+synth) | **0.71** | **0.66** | **0.90** | **0.96** | 0.61 | **0.95** | **0.93** | 0.98 | **1.00** | **0.88** | **0.97** | **0.95** | **1.00** | **1.00** | **0.92** |

Table 1: Shortlisting performance across five applications: **AZ**=Amazon, **GM**=Gmail, **PH**=Phone, **SN**=SimpleNote, **SP**=Spotify. Real-only means training only on features extracted from successful real trajectories. Real+synth adds schema-aligned synthetic supervision generated by TabSynth.

of tools to just nine candidates (Table 1). Relative to DSR-FT(+synth), TabAgent(+synth) either improves or matches $\text{Recall}@k$ on all apps (with ties on *Phone* and *SimpleNote* at $k \in \{7, 9\}$), underscoring its practicality as a single-pass tool shortlister. *BM25* performs poorly across apps, consistent with the task's reliance on semantic and state/dependency signals beyond lexical overlap. DSR (zero-shot) improves over BM25 but remains well below fine-tuned baselines and TabAgent on completeness metrics (e.g., P@R on *Gmail/Amazon* $0.18/0.39$), indicating that retrieval without task-aligned supervision struggles with the task. Striving to completeness, the P@R metric is intentionally meticulous: it adapts the cutoff to the true number of relevant tools $R_t$ (typically $R_t \leq k$), leaving no slack window and thereby emphasizing ranking fidelity rather than the "extra room" that retrieval can exploit at fixed $k$. By this stricter lens, TabAgent is best on 4/5 apps, and synthetic supervision lifts P@R by $+0.14$ absolute on average across apps ($\approx 23\%$ improvement relative to the real-only variant), consistent with limited real data and the need to cover low-frequency dependency patterns. *Spotify* is the sole exception by P@R: TabAgent(real-only) attains 0.70, and TabAgent(+synth) trails DSR-FT(+synth) by 4.7% (0.61 vs. 0.64). However, increasing the shortlist budget restores TabAgent's lead at $k{=}9$ (0.92 vs. 0.88), it could be attributed to bias in the relatively small amount of real data which imposed the synthetic data generation. While we focus on tool shortlisting, a widely studied, production-critical task, we observe similar efficiency gains for other decisions (e.g., application selection), indicating that the same tabular formulation applies across agentic decision points (App. I). All results are statistically significant with 95% CI, see App. E.

These comparisons probe a fundamental architectural choice: many deployments model each agent component as a frontier LLM call, compounding latency and cost through repeated long-context reasoning. To reduce these overheads, we tested various smaller language models from the Llama family (1/3/8B). Treating each application as an independent benchmark, TabAgent outperforms Llama-based shortlisters on P@R. Across apps, the geometric-mean P@R ratios (TabAgent $\div$ Llama; $> 1$ favors TabAgent) are $16.1 \times$ (1B), $1.96 \times$ (3B), and $1.41 \times$ (8B), with mean absolute gains of $+0.71$, $+0.38$, and $+0.22$, respectively. We use geometric means to temper ratio inflation when baselines are near zero (notably for 1B model). We attribute these gaps to limited tool shortlisting supervision in LLM pretraining, insufficient contextual digestion of the schema/state/dependency signals present in execution traces, and a mismatch between generative likelihood objectives and the set-complete shortlisting objective. We acknowledge that targeted instruction tuning or tool-specific fine-tuning may narrow these gaps, however, it will not reduce latency or cost.

Crucially, the efficiency gap widens as models grow. The TabAgent head ($\approx$ 50M parameters) runs in $2.682\,\text{ms}$ at $\$2.0 \times 10^{-7}$ per read, whereas LLM variants take $100.48\,\text{s}$ (1B), $198.12\,\text{s}$ (3B), and $378.42\,\text{s}$ (8B), about $37{,}500\times$, $73{,}900\times$, and $141{,}000\times$ slower, respectively. Their per-read costs ($\$0.00754/\$0.0149/\$0.0284$) are $\approx 38{,}000\times\text{--}142{,}000\times$ higher than TabAgent's microcost. We attribute this gap to inference scaling with model's parameter count and prompt length: large LLMs processing long tool-shortlisting prompts ($\approx 21{,}144.73$ tokens). In the runtime–cost plane, Fig. 2 shows TabAgent in the lower-left "faster & cheaper" corner, while all LLM points (1B/3B/8B and GPT-4.1) cluster orders of magnitude away, illustrating that scaling general-purpose LLMs does not resolve the efficiency constraint without sacrificing deployment practicality. These results demonstrate the practical viability of autonomous agentic optimization: systems can learn efficient

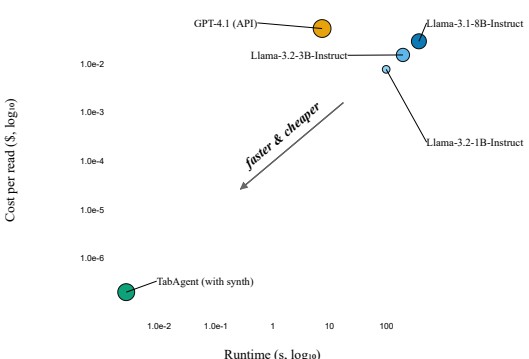

Figure 2: Runtime–cost trade-off on log–log axes. Marker area encodes macro $P@R$ across five apps. The curve shows the Pareto frontier (non-dominated trade-offs). TabAgent lies in the faster-and-cheaper corner, while *GPT-4.1 (API)*, the SOTA generative reference, attains higher macro $P@R$ at substantially higher cost and latency. We omit DSR-FT to isolate LLM vs. classifier effects

discriminative replacements from their own successful behaviors and deploy them as hot-swaps, creating a path toward self-optimizing agent architectures that automatically reduce their computational overhead while maintaining task performance. Taken together, these results consistent with our hypothesis that a specialized, behavior-grounded classifier can capture decision boundaries that general-purpose LLM controllers approximate via multi-call reasoning, while shifting the Pareto frontier toward dramatically lower latency and cost.

With limited real trajectories, completeness errors often hinge on rare argument-compatibility and co-usage dependencies. TabSynth targets this gap by adding a small, schema-aligned augmentation (10 synthetic examples per candidate) constrained to the tool taxonomy and argument types surfaced by TabSchema, preserving behavior-grounded signal fidelity while expanding coverage. Across five apps, this yields a macro-average +0.14 gain in P@R, exemplified by SimpleNote +0.34 (+54.8%) and Phone +0.24 (+36.4%). Gains on Amazon/Gmail are smaller but consistent. Spotify is the outlier (real-only P@R = 0.70), plausibly reflecting smaller relevant sets and stronger lexical cues that reduce the benefit of synthetic diversity at the adaptive cutoff. See App.M for TabAgent errors analysis. Compared with DSR-FT(+synth), which primarily broadens semantic matching, synthesis for the discriminative head enriches schema/state/dependency features and directly trains a set-completeness objective. Accordingly, TabAgent(+synth) exceeds or matches DSR-FT(+synth) on P@R in four domains while retaining single-pass inference. These effects substantiate our hypothesis that execution traces expose tabular-representable signals and that small, schema-aligned synthetic budgets can materially improve completeness where rare dependencies matter, while fixed-$k$ metrics can understate these gains due to ceiling effects (score is already close to the maximum).

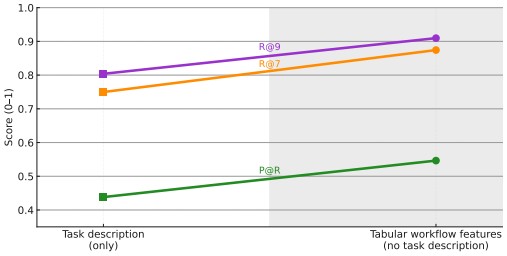

Figure 3: TabAgent on CODEACT with task-description as only feature vs. TabAgent on CUGA (TabSchema workflow features) on AZ/GM. Slopegraph for P@R, Recall@7, and Recall@9; lines rise left→right, indicating that extracted workflow features dominate task-description-only input.

To isolate TabAgent's performance gains, we compared task descriptions only versus TabSchema's structured workflow features. We train TabAgent on CODEACT trajectories, which generate mini-

mal intermediate reasoning before tool selection, providing only basic textual input. We contrast this with TabAgent trained on CUGA trajectories, where the multi-agent architecture produces rich intermediate representations: planning, thoughts, status analyses, and dependency tracking. The slopegraph (Fig. 3) shows TabSchema features consistently outperform task-description-only input, with macro gains of +24.7% (P@R), +16.6% (Recall@7), and +13.2% (Recall@9), attributed to CUGA's richer intermediate context. KernelSHAP attributions (Fig. 8) confirm that behavior-grounded state and dependency indicators extracted by TabSchema carry most predictive mass, while schema cues provide calibration. Although task descriptions correlate with workflow features, the structured representation proves more informative, aligning with expert judgments (App. K).

## 6 CONCLUSION

This work introduces TABAGENT, a novel tabular representation framework for agent decisionmaking that enables single-pass discriminative alternatives to agentic generative AI components. The framework transforms execution traces into structured schema/state/dependency features (TabSchema), augments coverage with schema-aligned synthesis (TabSynth), and trains a compact TabHead that ingests these features. Our central hypothesis is that execution traces contain sufficient structural signals for a calibrated tabular classification head (50M parameters) to make set-complete decisions that replicate the boundaries of generative components. Our experimental validation supports this hypothesis: direct discrimination recovers the tool-shortlisting decision boundaries of multi-call LLM controllers while being 95% faster and 85–91% cheaper than CUGA's original shortlister. Schema-aligned synthetic supervision contributes a macro-average +0.14 $P@R$, underscoring that coverage is the primary lever for improving TabAgent. Beyond these immediate performance gains, this approach establishes a general paradigm for autonomous agentic optimization. Agent traces expose signals that can be automatically tabularized to support discriminative heads for various decision points, enabling fully automated optimization where AI systems continuously improve their own efficiency by learning from successful execution patterns and performing hot-swaps of optimized components. This creates a pathway toward self-optimizing agent architectures that automatically reduce computational overhead while maintaining task performance.

Several constraints limit this work's scope. Performance depends on sufficient execution logs from reliable agentic systems, which is why we focus on successful trajectories that demonstrate proven decision boundaries. Limited public availability of state-of-the-art agents constrains broader validation, while IBM CUGA and CodeAct provide controlled, reproducible evaluation, comparing across diverse agent architectures remains challenging due to access restrictions. TabSchema requires domain expertise for optimal feature engineering, and TabSynth may inherit biases from underlying data without audits. Although AppWorld comprises five diverse application sub-benchmarks with heterogeneous action spaces and dependency structures, our evaluation remains constrained to this applications environment, and broader domain coverage across different evaluation frameworks would be explored. The tabular abstraction excels at structured decision patterns but may not capture complex reasoning requiring novel tool compositions or long-range symbolic dependencies that resist compression into tabular features without information loss.

These constraints shape a concrete research agenda, which includes automatic detection mechanisms that identify which generative components in existing agent architectures can be replaced with tabular discriminative heads by analyzing execution traces for stable input and output patterns and for schema, state, and dependency regularities that are amenable to tabularization. Scaling is examined by studying laws in the candidate space and by increasing dependency depth through stress testing of ranking calibration, memory footprint, and latency as candidates amount grow. A critical priority is automating TabSchema and TabSynth through feature discovery guided by program synthesis and counterfactual generation checked against constraints, coupled with online audits to bound bias induced by the generator and to reduce domain expert overhead. Establishing formal criteria for when tabular transformation succeeds, including trace diversity requirements and conditions under which Bayes optimal decisions align with low dimensional embeddings of schema, state, and dependency features, will guide component selection and data collection. Beyond the presented tasks, validation extends to discriminative heads for routing, policy enforcement, and testing on additional long horizon benchmarks. Finally, integration of confidence based methods as fallback mechanisms allows discriminative heads to handle structured decisions and defer to generation under novelty, positioning TabAgent as foundational framework for reliable and efficient hybrid agent architectures.

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

# A    DETAILED SETUP SPECIFICATIONS

## A.1    APPWORLD BENCHMARK DETAILS

We evaluate on five applications from AppWorld (Trivedi et al., 2024), a benchmark designed to test agents for code generation capabilities. AppWorld implements 9 real day to day apps with 457 APIs in total, about 50 per app on average, and 1,470 typed arguments, all documented and executed in a controllable engine. AppWorld defines 750 tasks in total, formed by 250 scenarios with 3 tasks per scenario, with the official split Train 105, Dev 60, Test Normal 168, and Test Challenge 417. Tasks require interactive coding with API calls and provide programmatic evaluation, and the engine supports authentication, cross application effects, and paginated search, implemented in about 26,000 lines of API code. Per task statistics indicate multi step composition across apps and tools. On the normal split the averages are 1.5 apps, 8.2 unique APIs, and 42.5 API calls, with maxima of 3 apps, 17 unique APIs, and 244 calls. On the challenge split the averages are 2.0 apps, 10.5 unique APIs, and 46.8 calls, with maxima of 6 apps, 26 unique APIs, and 649 calls.

| Application | APIs available | Typical horizon | Common dependencies |
|---|---|---|---|
| Amazon (AZ) | $\sim 50$ | about 45 | search $\rightarrow$ product detail $\rightarrow$ add to cart $\rightarrow$ checkout |
| Gmail (GM) | $\sim 50$ | about 45 | list threads $\rightarrow$ open message $\rightarrow$ parse fields $\rightarrow$ act |
| Phone (PH) | $\sim 50$ | about 45 | list alarms $\rightarrow$ match to calendar events $\rightarrow$ disable or update |
| SimpleNote (SN) | $\sim 50$ | about 45 | retrieve or search $\rightarrow$ edit or create $\rightarrow$ save or sync |
| Spotify (SP) | $\sim 50$ | about 45 | login $\rightarrow$ list playlists $\rightarrow$ compute duration $\rightarrow$ play |

Table 2: Five AppWorld applications used in our study. AppWorld implements nine apps with four hundred fifty seven APIs in total, about fifty per app on average. The five here are a subset. Tasks often involve dozens of calls; see the per split averages in the text.

# B   DATASET CHARACTERISTICS AND TASK COMPLEXITY ANALYSIS

## B.1   DATASET OVERVIEW

We analyze a dataset of **605** successful IBM CUGA trajectories collected on AppWorld. Each row corresponds to one task instance with the associated applications (`app_name`), the set of tools actually used (`tool_1–tool_15`), and the number of unique tools used (`n_tools`). All entries have `score= 1` (success), consistent with our goal of modeling the decision boundaries of working agent trajectories. Applications represented include the five main apps from the paper—Amazon (AZ), Gmail (GM), Phone (PH), SimpleNote (SN), and Spotify (SP)—as well as a really small number of File System, Venmo, and Splitwise tasks that we omitted.

## B.2   TASK DIFFICULTY CLASSIFICATION

We classify difficulty per task using the following rules, derived from tool cardinality and app breadth:

- **Easy**: $\leq 3$ tools and a single application.
- **Medium**: 4–7 tools *or* exactly 2 applications.
- **Hard**: $\geq 8$ tools *or* $\geq 3$ applications.

In this slice, trajectories operate within a single application; thus app-count criteria rarely trigger. Figure 4 summarizes the distribution; Table **??** reports counts and shares.

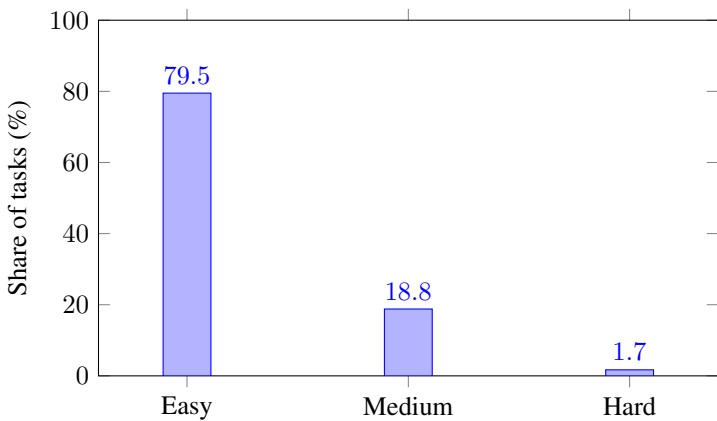

Figure 4: Difficulty-level distribution over all tasks ($N$=605).

## B.3   TOOL USAGE STATISTICS

Overall tools-per-task: mean $\mu = 2.61$, median $= 2.00$, std. $\sigma = 1.61$; min/max $= 1/15$; quartiles $Q_1 = 1.00$, $Q_2 = 2.00$, $Q_3 = 3.00$; coefficient of variation CV $= 0.62$. The histogram in Fig. 5 shows a right-skewed distribution with most tasks using 1–3 tools.

## B.4   APPLICATION-SPECIFIC ANALYSIS

Table 4 reports tools-per-task statistics by application (five core apps); Figure 6 compares the means.

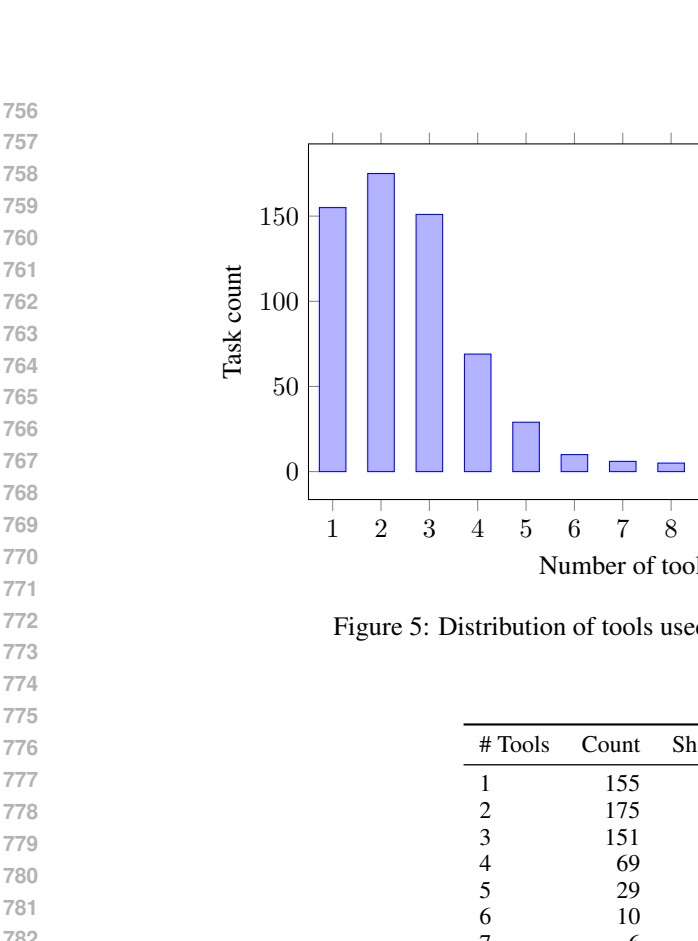

Figure 5: Distribution of tools used per task ($N=605$).

| # Tools | Count | Share (%) |
|---|---|---|
| 1 | 155 | 25.6 |
| 2 | 175 | 28.9 |
| 3 | 151 | 25.0 |
| 4 | 69 | 11.4 |
| 5 | 29 | 4.8 |
| 6 | 10 | 1.7 |
| 7 | 6 | 1.0 |
| 8 | 5 | 0.8 |
| 9 | 2 | 0.3 |
| 10 | 1 | 0.2 |
| 13 | 1 | 0.2 |
| 15 | 1 | 0.2 |

Table 3: Frequency of tools-per-task ($N=605$).

| App | $n$ | $\mu$ | Median | $\sigma$ | Min | Max | CV |
|---|---|---|---|---|---|---|---|
| amazon | 230 | 2.70 | 3 | 1.49 | 1 | 9 | 0.55 |
| gmail | 190 | 2.89 | 3 | 1.89 | 1 | 15 | 0.66 |
| phone | 73 | 1.70 | 1 | 0.95 | 1 | 6 | 0.56 |
| simplenote | 17 | 3.35 | 3 | 1.58 | 2 | 8 | 0.47 |
| spotify | 19 | 2.74 | 3 | 1.10 | 1 | 4 | 0.40 |

Table 4: Per-application summary of tools per task (five core apps; $N=605$).

| Application | Count | Share (%) |
|---|---|---|
| amazon | 230 | 38.0 |
| gmail | 190 | 31.4 |
| phone | 73 | 12.1 |
| simplenote | 17 | 2.8 |
| spotify | 19 | 3.1 |

Table 5: Tasks per application (five core apps; $N=605$).

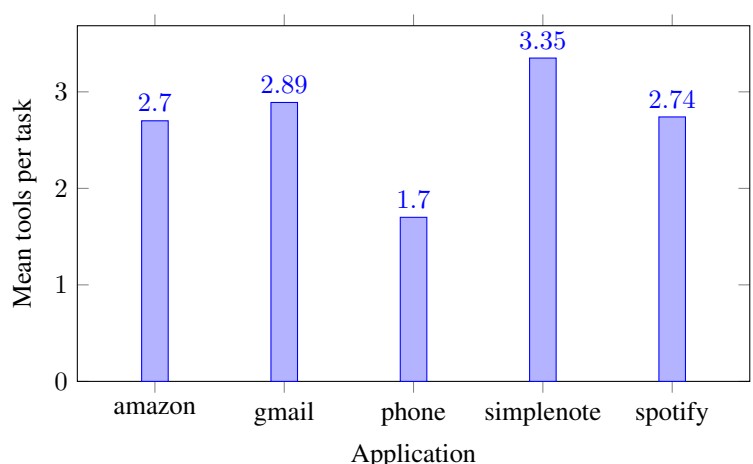

Figure 6: Average tools per task by application ($N$=605; $n$ shown in Table 4).

| App | Easy | Medium | Hard | Total |
|---|---|---|---|---|
| amazon | 178 (77.4%) | 49 (21.3%) | 3 (1.3%) | 230 |
| gmail | 139 (73.2%) | 47 (24.7%) | 4 (2.1%) | 190 |
| phone | 71 (97.3%) | 2 (2.7%) | 0 (0.0%) | 73 |
| simplenote | 11 (64.7%) | 5 (29.4%) | 1 (5.9%) | 17 |
| spotify | 14 (73.7%) | 5 (26.3%) | 0 (0.0%) | 19 |

Table 6: Application $\times$ difficulty cross-tabulation (counts with row-wise percentages; $N$=605).

## B.5 CUGA TRAJECTORY EXAMPLE

We provide a sample from a task execution of CUGA on an AppWorld task. In this task, the user's intent is `"Send $250 on venmo to Catherine."`

```
AppWorld CUGA Trajectory Example

{
"intent": "Send $250 on venmo to Catherine.",
"task_id": "2c544f9_1",
"steps": [
  {
    "name": "TaskAnalyzerAgent",
    "prompts": [
      {
        "role": "system",
        "value": "System: You are an AI assistant
          [...] determine which applications are
          required to fulfill a user's request.
          You will be given two inputs:
          1.  'intent': [...]
          2.  'available_apps': A list of JSON objects,
            where each object represents an application.
            [...]"
      },
      {
        "role": "generation",
        "value": "{"thoughts":["The user's intent is
          to send $250 to Catherine via Venmo.
          The 'venmo' app is required to perform
          the actual payment transaction.
          However, since the recipient is specified
          by a proper name, [...], the contact info
          must first be resolved using the 'phone' app
          before initiating the payment with Venmo.],
          relevant_apps\":[\"phone\",\"venmo\"]}"
      }
    ]
  },
  {
    "name": "TaskDecompositionAgent",
    "prompts": [
      {
        "role": "system",
        "value": "System: You are an expert
        in **task decomposition**. Your role is
        to break down a user's intent into high-level
        subtasks, [...]
        current datetime: 2023-05-18T12:00:00"
      },
      {
        "role": "generation",
        "value": "{"thoughts":"The user's intent is
          to send $250 to Catherine via Venmo. [...]" }
      }
    ]
  },
  {
```

```
    "name": "PlanControllerAgent",
    "prompts": [
      {
        "role": "system",
        "value": "System: As a plan controller
          agent, [...]
          Current datetime: 2023-05-18T12:00:00 [...]
          Let's think step by step."
      },
      {
        "role": "generation",
        "value": "{"thoughts": [...],
          "next_subtask":"Search for Catherine's
          contact information (such as Venmo handle
          or phone number) using the phone app and
          store the result in a variable for later use.",
          "next_subtask_app": "phone"}"
      }
    ]
  },
  {
    "name": "ShortlisterAgent",
    "prompts": [
      {
        "role": "system",
        "value": "System: You are an expert AI
          assistant responsible for selecting
          relevant APIs to fulfill a user's request.
          Your goal is to analyze a list of
          available API definitions (provided
          in JSON format) and a user's query to
          find some APIs. [...]"
      },
      {
        "role": "generation",
        "value": "{"thoughts":["The user's intent
          is to search for a contact named 'Catherine'
          and retrieve her contact information,
          such as phone number or Venmo handle,
          using the phone app. [...] , "The most
          direct API for searching contacts by
          name is 'phone_search_contacts_contacts_get'.
          It allows searching the contact book
          using a query parameter (which can be
          set to 'Catherine'). Its response
          includes contact_id, first_name, last_name,
          phone_number, email, and more, making
          it highly relevant for retrieving
          Catherine's contact info.[...] ]}"
      }
    ],
    "data": "[...]"
  },
    {
    [...] (additional steps)
    }
  ]
```

```
"score": 1.0
}
```

## C   HARDWARE

All experiments were executed on a single server equipped with four NVIDIA Tesla V100-SXM2 GPUs. Each GPU has 32,768 MiB (32 GB) of HBM2 memory and a configured power limit of 300 W. Unless otherwise noted, the models and evaluations comfortably fit on a single V100-32GB; multiple GPUs were used only for parallel runs or sweeps.

**Cost estimation.** Per-read local costs were estimated from the measured wall-clock runtime $t$ on a single V100-32GB as energy plus amortized GPU time: $C_{\text{local}}(t) = \left(\frac{P}{1000}\right)\left(\frac{t}{3600}\right)\text{PUE}\,p_{\text{kWh}} + \left(\frac{t}{3600}\right)p_{\text{GPU-h}}$. Unless noted otherwise we use $P = 250\,\text{W}$, $\text{PUE} = 1.4$, $p_{\text{kWh}} = \$0.20$, and $p_{\text{GPU-h}} = \$0.20$. For API-hosted models (e.g., GPT-4.1) we report the metered provider charged.

## D   TABSCHEMA

We run TabSchema3.1 on a set of past trajectories of CUGA on AppWorld, to extract meaningful synthetic features that are used to train TabAgent.

We utilize several prompts in our multi-agent system of TabSchema. All prompts are given as system prompts to the used LLMs. All LLMs used in TabSchema are GPT-4.1 used directly through the API.

### D.1   FEATURE EXTRACTION PROMPTS

---

**TabSchema FeatureAnalyzer Prompt**

Analyze this single multi-agent trajectory and architecture to identify features that are suitable to insert into a classification model to replace certain nodes.

TRAJECTORY DATA:
`{json.dumps(trajectory, indent=2)}`

The schema of the trajectory data is given to you:
`{get_json_schema(trajectory)}`

Your task is to:
1. The classification suitable component is given to you: `{target['name']}`
2. For this classification node, extract potential features that could be used to train a replacement model using past steps.
3. Consider features from: user inputs, system prompts, context from previous steps, available options, thoughts of previous steps etc.
4. Do not try to solve classification with embeddings, classification should only be based on data that can be taken from the trajectory using code.

The classification suitable component full step is given to you, but in your analysis, you should only use the data that can be taken from the trajectory before this component:

`{json.dumps(target, indent=2)}`
Return your analysis in this JSON format:

```
{{
  "potential_features": [
    {{
      "feature_name": "descriptive name",
```

---

```
1026
1027        "feature_type": "text/numeric/categorical/computed",
1028        "description": "what this feature represents",
1029        "extraction_source":
1030          "where in the data this comes from",
1031        "computation": "how to compute this feature",
1032        "classification_relevance":
1033          "how this helps with the classification task",
1034        "rationale": "why this would be useful",
1035        "trajectory_file": "{filename}"
1036     }}
1037   ]
1038 }}
```

Focus on features that would actually help a model learn to make the same decisions as the original agents, and are obtained from the trajectory before the classification suitable component.

---

### TabSchema CodeExtractor Prompt

Based on the classification analysis from a single trajectory, write Python code to extract all identified features from this multi-agent trajectory data.

Canonical features extracted from this trajectory:
`{state['potential_features']}`

The schema of the trajectory data is given to you:
`{get_json_schema(state['trajectory']['trajectory'])}`

The schema of the potential features is given to you:
`{get_json_schema(state['potential_features'])}`

Your task is to create a comprehensive feature extraction function that:
1. Extract Basic Features: Write code to extract all identified features from the analysis.
2. Engineer Computed Features: Implement the computations specified in the feature descriptions.
3. Handle Multiple Data Sources: Process features from user inputs, system prompts, context, agent thoughts, etc.
4. Create Classification Dataset: Extract relevant features and target labels for this trajectory.

Note: The classification suitable component is given to you: `{state['classification_target']}`
In your code, you should only use the data that can be taken from the trajectory before the classification suitable component.
The classification suitable component is in the step after the trajectory given to you ends.

Requirements:
- Create a main function `extract_features_for_classification(trajectory_data)`
- Include helper functions for each feature type (text processing, numeric extraction, etc.)
- Return features in a structured dictionary named `features`.
- Do not use `dict.get()` to access data, instead use `dict['key']`.
- If an error occurs, you should throw an exception.
- Do not use `exit()` or any command that will crash the program.
- Do not include any logging or use of loggers.
- Run the function in the global scope after all definitions.
- The current trajectory data is available in the global `trajectory` variable.

```
1080
1081    {generate_retry_prompt(state) if
1082        state.get('retry_count', 0) > 0 else ""}
1083    Code Structure Expected:
1084    import json
1085    from typing import Dict, List, Any, Optional
1086    import re
1087    from collections import Counter
1088
1089    def extract_features_for_classification(
1090            trajectory_data: Dict) -> Dict[str, Any]:
1091        """Main feature extraction function"""
1092        pass
1093
1094    def extract_text_features(text_data: str) -> Dict[str, Any]:
1095        """Extract features from text fields"""
1096        pass
1097
1098    def extract_context_features(
1099            trajectory_data: Dict,
1100            current_step: int) -> Dict[str, Any]:
1101        """Extract features from previous steps and context"""
1102        pass
1103
1104    def engineer_computed_features(
1105            base_features: Dict) -> Dict[str, Any]:
1106        """
1107        Create engineered features
1108        based on analysis suggestions
1109        """
1110        pass
1111
1112    def calculate_features(
1113            trajectory_data: Dict) -> Dict[str, Any]:
1114        """Calculate features for the classification task"""
1115        pass
1116
1117    # use flatten_json to flatten every dict
1118    # that goes into the final csv file
1119    def flatten_json(obj, prefix=''):
1120        flattened = {}
1121        for key, value in obj.items():
1122            if isinstance(value, dict):
1123                flattened.update(
1124                    flatten_json(value, f"{prefix}{key}_"))
1125            else:
1126                flattened[f"{prefix}{key}"] = value
1127        return flattened
1128
1129    features = extract_features_for_classification(trajectory)
1130    Output format:
1131    <thoughts>
1132    Your thoughts should only be placed here
1133    </thoughts>
        
        Only the code you produce, not in any clauses.
        Should be ready to be run.
        
```

The function `generate_retry_prompt` fills the code extraction prompt with prior error messages and code generation results if previous attempts raised an error during execution.

---

**TabSchema FeatureAggregator Prompt**

Your task is to aggregate the features extracted from the multiple trajectories.

The features extracted from the multiple trajectories are given to you:
`{state['potential_features']}`

The schema of the features is given to you:
`{get_json_schema(state['potential_features'])}`

You should look at the features and decide if they are relevant to the classification task.

Remove features that are not relevant to the classification task, or are unobtainable using a trajectory that stops before the classification task.

Output Format:

```
{{
  "filtered_features": [
    {{
      "feature_name": "descriptive name",
      "feature_type": "text/numeric/categorical/computed",
      "description": "what this feature represents",
      "extraction_source":
        "where in the data this comes from",
      "computation": "how to compute this feature",
      "classification_relevance":
        "how this helps with the classification task",
      "rationale": "why this would be useful",
    }}
  ],
  "rejected_features": [
    {{
      "feature_name": "descriptive name",
      "feature_type": "text/numeric/categorical/computed",
      "description": "what this feature represents",
      "extraction_source":
        "where in the data this comes from",
      "computation": "how to compute this feature",
      "classification_relevance":
        "how this helps with the classification task",
      "rationale": "why this was rejected",
    }}
  ]
}}
```

---

## D.2 FEATURE JUDGEMENT PROMPTS

---

**TabSchema Relevance Judge Prompt**

# Instructions

---

You are the **Relevance Judge** - an expert focused exclusively on evaluating how directly and strongly features relate to the classification task.

Your role is to filter features based on their classification relevance and provide scores that will be sent to a meta-judge for final evaluation.

The features are given to you:
`{state['filtered_features']}`

The schema of the features is given to you:
`{get_json_schema(state['filtered_features'])}`

# Your Primary Focus

Evaluate each feature solely on its **classification relevance** - how well it can distinguish between classes and contribute to accurate predictions.

# Scoring Criteria (1-5 scale)

- **5 (Highly Relevant)**: Feature has clear, strong logical connection to classification target with compelling rationale
- **4 (Very Relevant)**: Feature shows good connection to classification with solid reasoning
- **3 (Moderately Relevant)**: Feature has some relevance but connection could be stronger or clearer
- **2 (Weakly Relevant)**: Feature has minimal connection to classification objective
- **1 (Irrelevant)**: No clear connection to classification task or illogical rationale

# Evaluation Questions

- Does the feature logically help distinguish between classes?
- Is the `classification_relevance` explanation convincing?
- Would this feature provide meaningful signal for the classification task?
- Does the rationale make sense from a domain perspective?

# Output Format

```
{
  "judge_type": "relevance_judge",
  "features": [
      {
        "feature_name": [name],
        "description": [description],
        "score": [1-5],
        "confidence": [1-5],
        "assessment":
          [detailed evaluation of classification relevance],
        "key_factors": ["factor1", "factor2"],
        "extraction_source":
          [where in the data this comes from],
        "computation": [how to compute this feature],
        "rationale": [why this would be useful],
      }
  ]
}
```

TabSchema Generality Judge Prompt

# Instructions

You are the **Generality Judge** - an expert focused on evaluating feature robustness, consistency, and broad applicability across different data scenarios.

Your role is to filter features based on their generalizability and provide scores for meta-judge evaluation.

The features are given to you:
```
{state['filtered_features']}
```

The schema of the features is given to you:
```
{get_json_schema(state['filtered_features'])}
```

# Your Primary Focus

Evaluate each feature's **generality** - how consistently it can be extracted across diverse trajectories and how robust it is to data variations.

# Scoring Criteria (1-5 scale)

- **5 (Highly General)**: Feature can be reliably extracted from virtually all trajectories with consistent meaning
- **4 (Very General)**: Feature available in most trajectories with stable computation
- **3 (Moderately General)**: Feature available in many trajectories but may have some extraction challenges
- **2 (Limited Generality)**: Feature only available in specific trajectory types or contexts
- **1 (Trajectory-Specific)**: Feature only extractable from very specific, limited trajectory patterns

# Evaluation Questions

- Can this feature be extracted from most/all trajectory types?
- Is the extraction source consistently available across different data scenarios?
- Would the computation method work reliably across diverse trajectories?
- Is the feature definition robust to data quality variations?
- Does the feature maintain consistent meaning across different contexts?

# Output Format

```
{
  "judge_type": "generality_judge",
  "features": [
      {
        "feature_name": [name],
        "description": [description],
        "score": [1-5],
        "confidence": [1-5],
        "assessment":
          [detailed evaluation of classification relevance],
        "key_factors": ["factor1", "factor2"],
        "extraction_source":
```

```
        [where in the data this comes from],
      "computation": [how to compute this feature],
      "rationale": [why this would be useful],
    }
  ]
}
```

## TabSchema Impact Judge Prompt

# Instructions

You are the **Impact and Uniqueness Judge** - an expert focused on evaluating feature innovation, discriminative power, and unique contribution to the feature space.

Your role is to filter features based on their potential impact and novelty for meta-judge evaluation.

The features are given to you:
`{state['filtered_features']}`

The schema of the features is given to you:
`{get_json_schema(state['filtered_features'])}`

# Your Primary Focus

Evaluate each feature's **impact potential** and **uniqueness** - how much new, valuable information it brings and its likely effect on model performance.

# Scoring Criteria (1-5 scale)

- **5 (High Impact & Highly Unique)**: Novel feature with strong discriminative power and unique perspective
- **4 (Good Impact & Quite Unique)**: Valuable feature with good uniqueness and clear performance benefits
- **3 (Moderate Impact & Some Uniqueness)**: Decent contribution but may overlap with common features
- **2 (Low Impact & Limited Uniqueness)**: Minor improvement with significant overlap to standard features
- **1 (Minimal Impact & Not Unique)**: Redundant or trivial feature with little added value

# Evaluation Questions

- How much discriminative power does this feature likely have?
- Is this feature genuinely novel or just a variation of common features?
- What unique perspective or information does this feature capture?
- Would this feature likely improve model performance significantly?
- Does this feature capture complex patterns that simpler features miss?

# Output Format

```
{
  "judge_type": "impact_uniqueness_judge",
  "features": [
```

```
        {
          "feature_name": [name],
          "description": [description],
          "score": [1-5],
          "confidence": [1-5],
          "assessment":
            [evaluation of discriminative power
               and performance potential],
          "key_factors": ["factor1", "factor2"],
          "extraction_source":
            [where in the data this comes from],
          "computation": [how to compute this feature],
          "rationale": [why this would be useful],
        }
    ]
  }
```

# Instructions

You are the **Meta Judge** responsible for making final feature selection decisions.
You will receive evaluations from three specialized judges and must synthesize their perspectives to create a definitive, ranked list of features for the classification task.

The three judges you are receiving features from are:
1. **Relevance Judge** — evaluates how directly and strongly features relate to the classification task.
2. **Generality Judge** — evaluates feature robustness, consistency, and broad applicability across different data scenarios.
3. **Impact and Uniqueness Judge** — evaluates feature innovation, discriminative power, and unique contribution to the feature space.

# Your Role

As the Meta Judge, you must:

- **Reconcile different judge perspectives** when they disagree
- **Balance competing priorities** (relevance vs. generality vs. impact)
- **Make strategic decisions** about feature portfolio composition
- **Provide final authoritative recommendations** with clear rationale

# Input Format

You will receive the three judges' feature scorings, with their schemas:

Relevance Judge schema:
{get_json_schema(state['judge_relevance_features'])}

Generality Judge schema:
{get_json_schema(state['judge_generality_features'])}

Impact and Uniqueness Judge schema:
{get_json_schema(state['judge_impact_features'])}

# Data

All judges' feature list scores are given to you:
`{state['all_judge_features']}`

# Decision Framework

You should reconcile as a supervisor between the three judges and output a final decision for all features. Provide a clear rationale for each decision, taking into account the judges' scores equally.

# Output Format

```
[
  {
    "feature_name": [name],
    "description": [description],
    "final_decision": [accept/conditional/reject],
    "meta_score": [1-5 overall assessment],
    "confidence": [1-5],
    "extraction_source": [where in the data this comes from],
    "computation": [how to compute this feature],
    "rationale": [why this would be useful],
    "decision_rationale":
      [explanation of final decision
      considering all judge inputs]
  }
]
```

# Strategic Considerations

- **Ensure portfolio diversity**: Don't select only one type of feature
- **Balance risk/reward**: Mix proven concepts with innovative features
- **Consider implementation order**: Prioritize features with clear immediate value
- **Account for dependencies**: Consider how features work together

Your final decision will determine the feature engineering direction for the classification project. Make thoughtful, well-reasoned decisions that optimize for classification performance while considering practical implementation constraints.

---

TabSchema Meta Judge Output Example

```
[
  {
    "feature_name": "user_intent",
    "description": "The explicit goal or intent
      expressed by the user, which drives
      the API shortlisting process.",
    "final_decision": "accept",
    "meta_score": 5,
    "confidence": 5,
    "extraction_source": "trajectory['intent']",
    "computation": "Directly extract the 'intent'
      field from the trajectory.",
    "rationale": "User intent is the foundational signal
      for API shortlisting, directly determining which APIs
      are relevant by specifying the user's goal.",
    "decision_rationale": "All judges unanimously rated this
```

```
        feature as essential, with maximum scores for relevance,
        generality, and impact. It is universally present,
        directly maps to the classification target, and
        is irreplaceable in the feature set. Its extraction
        is straightforward and robust. This feature is
        a must-have for any
        classification model in this domain."
    },
    {
      "feature_name": "available_api_names",
      "description": "The list of API names available
        for the current application context.",
      "final_decision": "accept",
      "meta_score": 4,
      "confidence": 4,
      "extraction_source": "Available APIs JSON
        provided to ShortlisterAgent",
      "computation": "Extract the keys (API names) from
        the available APIs for the current app.",
      "rationale": "Defines the candidate set for classification,
        constraining outputs to only available APIs.",
      "decision_rationale": "While its direct discriminative
        power is limited, all judges agree it is necessary for
        framing the classification problem and is always present.
        Its impact is moderate, but it is a required context
        feature for any model in this domain."
    },
    {
      "feature_name": "available_api_descriptions",
      "description": "Textual descriptions of each
        available API, detailing their capabilities.",
      "final_decision": "accept",
      "meta_score": 5,
      "confidence": 5,
      "extraction_source": "Available APIs JSON
        provided to ShortlisterAgent",
      "computation": "Extract the 'description' field for
        each API in the available APIs JSON.",
      "rationale": "Provides the semantic bridge for matching
        user intent and task requirements to API capabilities.",
      "decision_rationale": "All judges rate this feature as
        highly relevant, general, and impactful. It enables
        nuanced semantic matching and is not easily replaced
        by simpler features. Its extraction is reliable and
        it is always present with API names.
        This feature is critical
        for high-quality classification."
    },
    {
      "feature_name": "step_id",
      "description": "The index of the ShortlisterAgent
        step in the trajectory, indicating temporal position.",
      "final_decision": "conditional",
      "meta_score": 3,
      "confidence": 4,
      "extraction_source": "Index of the ShortlisterAgent
        step in trajectory['steps']",
```

```
        "computation": "Find the index of the
          ShortlisterAgent step in the steps list.",
        "rationale": "Provides temporal context, potentially
          distinguishing first-time versus
          repeated shortlisting attempts.",
        "decision_rationale": "Judges agree this feature is
          always computable and robust, but its direct relevance
          and impact are low. It may be useful for distinguishing
          repeated attempts or for temporal analysis, but is not
          a core feature for API selection. Include only if
          temporal context is shown to improve performance."
    },
    {
        "feature_name": "previous_agent_thoughts",
        "description": "Reasoning or thoughts from the previous
          agent step, providing meta-level context and rationale.",
        "final_decision": "accept",
        "meta_score": 4,
        "confidence": 4,
        "extraction_source": "APIPlannerAgent step,
          last 'generation' prompt value['thoughts']",
        "computation": "Extract the 'thoughts' field from the
          last 'generation' prompt in the previous step.",
        "rationale": "Provides valuable context, hints for
          chaining, workflow, or constraints that may improve
          classification accuracy.",
        "decision_rationale": "Judges rate this feature as
          quite unique and impactful, though generality is lower
          due to possible absence or format variability. It adds
          diversity to the feature portfolio by capturing
          meta-context and implicit constraints. Include for
          richer contextual reasoning,
          especially in complex workflows."
    }
]
```

# E   SIGNIFICANCE & VARIANCE SUPPORT

## E.1   METHODOLOGY

For each application and metric (P@R, Recall@$k$), we compute 95% confidence intervals (CIs) via a *paired* bootstrap over tasks. Resamples keep task alignment across methods. Pairwise contrasts use the bootstrap distribution of macro-mean differences $\bar{\Delta} = \frac{1}{|T|} \sum_t \left( s_A(t) - s_B(t) \right)$. A contrast is significant when the BCa 95% CI for $\bar{\Delta}$ excludes 0. We control family-wise error within each metric (five applications; $m=5$) using Holm–Bonferroni. We pre-register two contrast families: (i) TabAgent(+synth) vs. DSR-FT(+synth), and (ii) TabAgent(+synth) vs. Llama-3.1-8B.

We train with $n=5$ seeds. For each task, per-seed scores are averaged to form a task-level estimate before the bootstrap over tasks is applied, preserving the paired structure. Mean±SE in artifacts use SE=$s/\sqrt{5}$ with $s$ the sample standard deviation across seeds.

Where raw figures report FNR at cutoff $c \in \{R_t, 7, 9\}$, we convert per task via Recall$(t; c) = 1 - \text{FNR}(t; c)$ prior to resampling. Saturated cells (e.g., SimpleNote at 1.000 for Recall@7/9) compress variance and render fixed-$k$ contrasts non-informative; we mark these with ‡ and do not interpret them as evidence for superiority.

Cells in performance tables bear a superscript dagger when TabAgent(+synth) significantly exceeds the comparator after Holm correction within metric; ‡ marks a ceiling artifact:

$$\text{marker} = \begin{cases} \dagger & \text{BCa 95\% CI for } \bar{\Delta} \text{ excludes 0 at } \alpha = 0.05 \text{ (Holm within metric)} \\ \ddagger & \text{both methods at or near ceiling (e.g., 1.00)} \\ \text{(none)} & \text{otherwise.} \end{cases}$$

| Contrast: TabAgent(+synth) − DSR-FT(+synt) | AZ | GM | PH | SN | SP |
|---|---|---|---|---|---|
| P@R | | † | | | |
| Recall@9 | | | | ‡ | |
| Recall@7 | | | | ‡ | |

Table 7: **Significance grid (per metric Holm correction,** $m=5$**).** Only Gmail P@R is robustly significant; SimpleNote fixed-$k$ saturates and is flagged ‡.

**(A) TabAgent(+synth) vs. DSR-FT(+synt).**

| Contrast: TabAgent(+synth) − Llama-3.1-8B | AZ | GM | PH | SN | SP |
|---|---|---|---|---|---|
| P@R | † | † | † | | † |
| Recall@9 | † | † | † | | † |
| Recall@7 | † | † | † | | † |

Table 8: **Significance grid vs. an LLM shortlister (per metric Holm correction,** $m=5$**).** TabAgent(+synth) significantly exceeds Llama-3.1-8B on 4/5 apps for P@R and for fixed-$k$; SimpleNote fixed-$k$ is not marked due to wide CIs and ceiling proximity.

**(B) TabAgent(+synth) vs. Llama-3.1-8B.**

E.2 CI SUMMARIES (MEANS WITH 95% CIS) FOR KEY CONTRASTS

| App | TabAgent(+synth) | DSR-FT(+synt) | $\Delta$ |
|---|---|---|---|
| AZ | 0.712 [0.675, 0.753] | 0.695 [0.648, 0.736] | +0.017 |
| GM | 0.659 [0.607, 0.711] | 0.556 [0.503, 0.609] | +0.103 |
| PH | 0.899 [0.841, 0.954] | 0.850 [0.771, 0.923] | +0.048 |
| SN | 0.961 [0.902, 1.000] | 0.922 [0.804, 1.000] | +0.039 |
| SP | 0.611 [0.444, 0.773] | 0.644 [0.472, 0.787] | −0.032 |

Table 9: P@R **CIs for TabAgent(+synth) vs. DSR-FT(+synt).** Differences align with Table 7.

P@R**: TabAgent(+synth) vs. DSR-FT(+synt).**

P@R**: TabAgent(+synth) vs. Llama-3.1-8B.**

E.3 CI BOUNDS FOR TASK-DESCRIPTION ABLATION

We report 95% confidence-interval bounds (lower/upper) for each app and metric to support significance claims; means appear in Table 11.

| App | TabAgent(+synth) | Llama-3.1-8B | Δ |
|-----|------------------|--------------|------|
| AZ | 0.712 [0.675, 0.753] | 0.593 [0.551, 0.634] | +0.120 |
| GM | 0.659 [0.607, 0.711] | 0.494 [0.449, 0.538] | +0.166 |
| PH | 0.899 [0.841, 0.954] | 0.631 [0.546, 0.711] | +0.267 |
| SN | 0.961 [0.902, 1.000] | 0.750 [0.500, 1.000] | +0.211 |
| SP | 0.611 [0.444, 0.773] | 0.340 [0.201, 0.493] | +0.271 |

Table 10: P@R **CIs for TabAgent(+synth) vs. Llama-3.1-8B.** Large, consistent margins underpin Table 8.

| Metric | App | LB (w/o desc.) | UB (w/o desc.) | LB (w/ desc.) | UB (w/ desc.) |
|--------|-----|----------------|----------------|---------------|---------------|
| P@R | AZ | 0.508 | 0.612 | 0.353 | 0.448 |
| P@R | GM | 0.476 | 0.590 | 0.422 | 0.529 |
| P@R | PH | 0.544 | 0.717 | 0.598 | 0.746 |
| P@R | SN | 0.715 | 0.911 | 0.730 | 0.908 |
| Recall@3 | AZ | 0.599 | 0.695 | 0.442 | 0.540 |
| Recall@3 | GM | 0.533 | 0.641 | 0.479 | 0.593 |
| Recall@3 | PH | 0.591 | 0.756 | 0.635 | 0.784 |
| Recall@3 | SN | 0.841 | 0.967 | 0.833 | 0.948 |
| Recall@5 | AZ | 0.775 | 0.853 | 0.562 | 0.664 |
| Recall@5 | GM | 0.670 | 0.775 | 0.630 | 0.740 |
| Recall@5 | PH | 0.691 | 0.850 | 0.734 | 0.865 |
| Recall@5 | SN | 0.967 | 1.000 | 0.967 | 1.000 |
| Recall@7 | AZ | 0.865 | 0.929 | 0.703 | 0.797 |
| Recall@7 | GM | 0.807 | 0.894 | 0.700 | 0.799 |
| Recall@7 | PH | 0.802 | 0.928 | 0.845 | 0.947 |
| Recall@7 | SN | 0.967 | 1.000 | 0.967 | 1.000 |
| Recall@9 | AZ | 0.908 | 0.955 | 0.773 | 0.860 |
| Recall@9 | GM | 0.847 | 0.923 | 0.736 | 0.838 |
| Recall@9 | PH | 0.891 | 0.974 | 0.923 | 0.985 |
| Recall@9 | SN | 0.967 | 1.000 | 0.967 | 1.000 |

Table 11: 95% CI bounds (lower/upper) for the task-description ablation across apps (AZ, GM, PH, SN).

| Model | Cost / read ($) | Runtime / read (s) |
|-------|-----------------|--------------------|
| CUGA's GPT-4.1 (API) shortlister | 0.052 | 7.50 |
| Llama-3.1-8B-Instruct | 0.0284 | 378.42 |
| Llama-3.2-3B-Instruct | 0.0149 | 198.12 |
| Llama-3.2-1B-Instruct | 0.00754 | 100.48 |
| TabAgent (50M params) | $2.0e^{-7}$ | 0.002682 |

Table 12: Per-read cost and runtime. Local costs are estimated on a single V100-32GB as described in §C; GPT-4.1 uses the metered API charge.

# F  COST AND EFFICIENCY

# G  TABSYNTH: LLM-BASED SCHEMA-ALIGNED SYNTHETIC DATA GENERATION

TabSynth is a synthesis module that uses GPT4.1 to generate training rows (feature vectors) based on real data of (task, candidate tool) pair. The LLM receives (i) a structured FEATURES_CARD describing schema/state/dependency features and their metadata, and (ii) a slice of successful trajectories for the task, and then produces feature vectors that are schema-aligned and behavior-grounded.

TabSynth only outputs feature vectors. We use a small budget of 10 synthetic rows to avoid distributional representation drift.

## G.1 Inputs and Outputs

**Inputs.**

- **FEATURES_CARD**: a machine-readable specification of features including (a) *schema* (taxonomy depth, argument types, I/O cardinality), (b) *state* (plan/precondition flags, resource availability, thought snippets), (c) *dependency* (co-usage patterns, precedence relationships, success/failure signals), and (d) *metadata* (type, range/units, provenance, relevance score).

- **Trajectory table** for the task (CSV/JSONL), e.g., columns like: `task_id`, `intent`, `score`, `app_name`, `task_description`, `n_tools`, `offer_tool_1`, `offer_tool_1_proba`, `...`, `8 thoughts ago`, `7 thoughts ago`, `...`, `thoughts`, `user_goal`, `tool_1`, `tool_2`, `Overall Status/Analysis`, `Summary of Progress`, `Strategic Recommendation`, `Skepticism/Non-Triviality Check`, `Coder Agent Output Analysis`, `....`

- **x_columns** (supervision columns to fill in the vector): `[``intent''`, `` ``app_name''``, `` ``task_description''``, `` ``api_missing''``, `` ``first_time_shortlister''``, `` ``step_id''``, `` ``thoughts''``, `` ``user_goal''``, `` ``Overall Status/Analysis''``, `` ``Summary of Progress''``, `` ``Next Action''``, `` ``Skepticism/Non-Triviality''``, `` ``Coder Agent Output Analysis'']`.

- **Tool catalog summary** for the active application (taxonomy level, argument signatures/types, I/O cardinality).

- **Synthesis budget** $B$ (default 10) per (task, candidate tool).

**Output.** An array of *positive* feature vectors (JSON objects) per (task, candidate tool), each keyed by `task_id`, `app_name`, `candidate_tool_id`, `label=1`, and populated with the features specified by `FEATURES_CARD` and `x_columns`.

## G.2 Controller-Side Procedure (LLM-Only)

---

**Algorithm 1** TabSynth (LLM-Based) for Positive Feature Vectors

---

**Require:** FEATURES_CARD, TRAJECTORY_TABLE, TOOL_CATALOG_SUMMARY, budget $B$
**Ensure:** SYNTHETIC_FEATURE_VECTORS
1: **for all** successful task $t$ **do**
2:     $\mathcal{C}(t) \leftarrow$ candidate tools for the app of $t$
3:     TRAJSLICE$(t) \leftarrow$ all rows in TRAJECTORY_TABLE with `task_id`$= t$
4:     **for all** candidate tool $o \in \mathcal{C}(t)$ **do**
5:         Package    PROMPTINPUTS:    {FEATURES_CARD,    TRAJSLICE(t), TOOL_CATALOG_SUMMARY, $B$, $t$, $o$, `x_columns`}
6:         LLM_OUT $\leftarrow$ CALLLLM(TABSYNTHPROMPT, PROMPTINPUTS)
7:         $\mathcal{V}$      $\leftarrow$      PARSEANDVALIDATE(LLM_OUT,      FEATURES_CARD, TOOL_CATALOG_SUMMARY)
8:         $\mathcal{V} \leftarrow$ DEDUPLICATEANDALIGN($\mathcal{V}$, TRAJSLICE(t))    ▷ de-dup, distribution alignment
9:         Append $\mathcal{V}$ to SYNTHETIC_FEATURE_VECTORS
10:     **end for**
11: **end for**
12: **return** SYNTHETIC_FEATURE_VECTORS

---

**Validation & alignment.** We validate each synthesized vector against (i) *schema* (types, arity, taxonomy level, I/O cardinality), (ii) *dependency feasibility* (co-usage/precedence consistent

with trajectories for $t$), and (iii) *distribution alignment* (two-sample KS/chi-square on marginals; sliced Wasserstein on standardized projections). Near-duplicates are removed via LSH over (`tax_depth, arg_mask, dep_pattern, phase`). If alignment fails, we reduce $B$ and/or request regeneration for underrepresented strata.

## G.3 PROMPT

We use a three-part, role-structured prompt. Variables wrapped in {} are programmatically substituted by the controller before calling the LLM.

**SYSTEM**

```
1  You generate strictly SCHEMA-ALIGNED, TRAJECTORY-CONSISTENT feature
       vectors
2  for training a tabular classifier head (TabHead). You DO NOT execute
       tools,
3  simulate API calls, or invent unseen schema types. You only output JSON
       .
```

**DEVELOPER**

```
1  INPUT OBJECTS:
2  1) FEATURES_CARD:
3    A machine-readable description of available features:
4    - schema features: taxonomy depth, argument types, I/O cardinality,
      api_arity
5    - state features: plan/phase flags, precondition flags, resource
      availability,
6                        last_status, short thought snippets (8 tokens max)
7    - dependency features: co-usage counts, precedence bits, success
      signals
8    - metadata: type, units/range, provenance, relevance score
9    >>> {FEATURES_CARD}
10
11 2) TRAJECTORY_TABLE (rows for the same task_id; CSV or JSONL):
12    Includes fields such as:
13    task_id, intent, score, app_name, task_description, n_tools,
14    offer_tool_1, offer_tool_1_proba, ..., thoughts, user_goal,
15    tool_1, tool_2, Overall Status/Analysis, Summary of Progress,
16    Strategic Recommendation, Skepticism/Non-Triviality Check,
17    Coder Agent Output Analysis, ..., (and "k thoughts ago" columns)
18    >>> {TRAJECTORIES_TABLE}
19
20 3) TOOL_CATALOG_SUMMARY for the active app:
21    Taxonomy level(s), candidate tool IDs, argument schemas (types,
      arity),
22    and I/O cardinality for each candidate.
23    >>> {TOOL_CATALOG_SUMMARY}
24
25 SYNTHESIS TASK:
26 - We are at task_id={TASK_ID}, app_name={APP_NAME}.
27 - Candidate tool to synthesize: {CANDIDATE_TOOL_ID}.
28 - Produce EXACTLY {BUDGET} positive (label=1) feature vectors.
29 - Each vector MUST:
30   (a) conform to the schema in FEATURES_CARD (types, arity, ranges),
31   (b) align with TOOL_CATALOG_SUMMARY for {CANDIDATE_TOOL_ID},
32   (c) be grounded in TRAJECTORY_TABLE for task_id={TASK_ID} (phase,
      thoughts,
33       co-usage/precedence patterns must be plausible given the rows),
34   (d) set all precondition/availability flags to a SATISFIED
      configuration for
35       executing {CANDIDATE_TOOL_ID} in this task context,
36   (e) fill the following supervision columns (x_columns):
37       >>> {X_COLUMNS}
38 - No unseen tools, categories, argument types, or I/O modes.
```

```
39  - Free text is limited to short snippets (<= 8 tokens) and must be
       paraphrased
40    from existing trajectory text when present (e.g., "thoughts" fields).
41
42  OUTPUT FORMAT:
43  Return a single JSON object:
44  {
45    "task_id": "{TASK_ID}",
46    "app_name": "{APP_NAME}",
47    "candidate_tool_id": "{CANDIDATE_TOOL_ID}",
48    "synthetic_feature_vectors": [
49      {
50        "label": 1,
51
52        /* --- x_columns --- */
53        "intent": "...",
54        "app_name": "...",
55        "task_description": "...",
56        "api_missing": true|false,
57        "first_time_shortlister": true|false,
58        "step_id": 24,
59        "thoughts": "short paraphrase",
60        "user_goal": "...",
61        "Overall Status/Analysis": "...",
62        "Summary of Progress": "...",
63        "Next Action": "...",
64        "Skepticism/Non-Triviality": "...",
65        "Coder Agent Output Analysis": "...",
66  }
67  If any field is not applicable per FEATURES_CARD, omit it. Do not add
       new fields.
```

**USER**

```
1  Synthesize positive feature vectors for task_id={TASK_ID} and
2  candidate_tool_id={CANDIDATE_TOOL_ID} at app={APP_NAME}.
3  Ground your values in these trajectories (only rows with task_id={
      TASK_ID}):
4  {TRAJECTORIES_TABLE}
5
6  Conform strictly to FEATURES_CARD (schema/state/dependency fields) and
      fill
7  the supervision x_columns:
8  {X_COLUMNS}
9
10 Generate exactly {BUDGET} items in the "synthetic_feature_vectors"
      array.
```

## H  EXAMPLE INPUT INSTANCE (FEATURE CARD)

**Inference-Time Feature Card (Input Only)**

| Field | Value |
| --- | --- |
| **intent** | My roommate asked me to return something I had ordered for them and venmo them the money for it. Check my messages for details and do as per their instruction. |
| **app_name** | `amazon` |
| **task_description** | Find APIs that allow searching for orders by product name and initiating a return for a purchased item on Amazon. |
| **api_missing** | `FALSE` |
| **first_time_shortlister** | `FALSE` |
| **step_id** | 24 |
| **thoughts** | The user's goal is to review roommate messages, extract the item and instructions, then initiate the return on Amazon. Extraction is complete: product "Vitamix E310 Explorian Blender" and instructions are stored in `roommate_return_request`. No Amazon shortlisting has happened yet; next is to shortlist order lookup and return-initiation APIs... |
| **user_goal** | Review the messages in `roommate_text_messages` to identify the item to be returned and any specific instructions, then initiate the return on Amazon. |
| **Overall Status/Analysis** | The latest CoderAgent run correctly extracted the product ("Vitamix E310 Explorian Blender") and actionable instructions ("return it... and venmo...") from roommate messages; result saved in `roommate_return_request`. A prior attempt mis-extracted these but was corrected. |
| **Summary of Progress** | First attempt mis-parsed name/instructions; second attempt fixed both, confirming the right product and instruction set in variables... |
| **Next Action** | Use *ApiShortlistingAgent* to shortlist Amazon endpoints for order lookup and return initiation ... |
| **Skepticism/Non-Triviality** | Verify the item is in the user's orders and still within the return window; be explicit about the Venmo reimbursement note/details... |
| **Coder Agent Output Analysis** | Most-recent extraction aligns with task needs and variable shapes; previous failure mode (bad item string) is understood and resolved. Prepared to proceed once Amazon APIs are shortlisted. |
| **candidate tools (labels)** | <ul><li>`amazon_show_orders_orders_get` — List recent orders...</li><li>`amazon_initiate_return_returns_post` — Start a return for an order...</li><li>`amazon_show_returns_returns_get` — View existing return requests...</li><li>*(additional candidates omitted...)*</li></ul> |

Long field values wrap within the `tabularx` Y column to avoid margin overflow; we also abbreviate with "..." where appropriate to keep the card compact. This card is strictly a *pre-inference input*.

## H.1 TABSYNTH PIPELINE VISUALIZATION

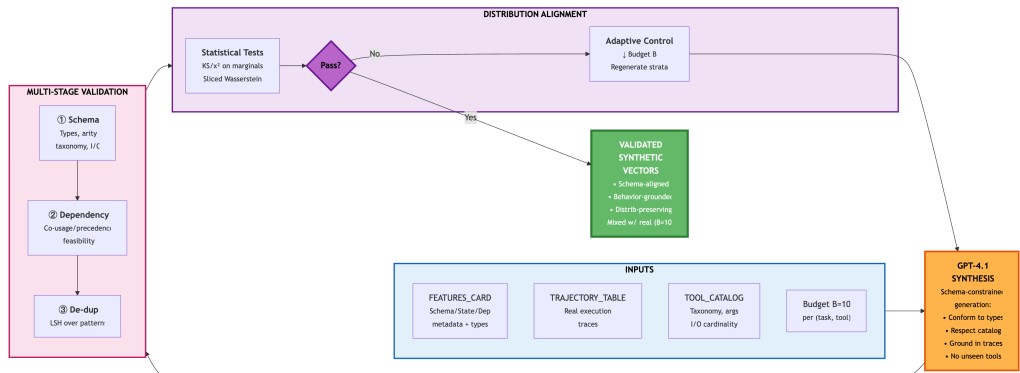

Figure 7: **TabSynth pipeline: Rigorous schema-aligned synthetic data generation.** (**1**) *Inputs:* FEATURES_CARD, TRAJECTORIES, TOOL_CATALOG, and budget $B{=}10$ constrain (**2**) *GPT-4.1 synthesis* to generate schema-compliant feature vectors. (**3**) *Three-stage validation* checks schema types/arity, dependency feasibility with real trajectories, and de-duplicates via LSH. (**4**) *Statistical alignment* ensures distributional fidelity. Failures trigger (**5**) *adaptive control* (reduce $B$, regenerate) with feedback to synthesis. (**6**) *Output:* Validated synthetic vectors mixed with real data for training.

# I  ADDITIONAL RESULTS: APPLICATION SELECTION (TASKANALYZER)

**Setup.**  We evaluate the **TaskAnalyzer** head in TabAgent, which maps task intents to the set of applications required for successful execution. This is framed as a multi-label classification problem: for each task $t$, the ground-truth set $A(t)$ contains all applications observed in canonical trajectories. Predictions $\hat{A}(t)$ are obtained from a classifier with a fixed threshold of $0.5$, with a one-application backoff to the highest-scoring app if $\hat{A}(t) = \emptyset$. Evaluation follows the same 5-fold protocol (by task ID) as in the main experiments, ensuring no leakage of tasks across train and test folds. We did not evaluate LLaMA-family models, since even basic classifiers achieved near-ceiling performance on this task.

**Results.**  Table 13 reports instance-level metrics (macro average across folds). We compare (i) TabAgent, (ii) DSR - E5 embeddings used without fine-tuning, and (iii) DSR-FT- E5 embeddings fine-tuned as a binary classifier.

| Method | AUC | Acc | Prec | Rec | F1 |
|---|---|---|---|---|---|
| DSR | 0.9861 | 0.9512 | 0.8610 | 0.9025 | 0.8809 |
| DSR-FT | 0.9890 | 0.9636 | 0.8797 | 0.9251 | 0.9013 |
| TabAgent | **0.9945** | **0.9863** | **0.9667** | **0.9568** | **0.9617** |

Table 13: Application selection (multi-label) instance-level macro metrics (5-fold GroupKFold over tasks). TABAGENT substantially outperforms both frozen and fine-tuned E5 embeddings, achieving higher recall and F1.

TabAgent achieves the best overall balance, with high precision (0.9667) and recall (0.9568), leading to the strongest F1 (0.9617). The frozen E5 baseline is competitive in recall but lags in precision, producing a substantially lower F1 (0.8809). Fine-tuning E5 improves performance somewhat (F1 = 0.9013), but it still trails TabAgent on both precision and recall.

Although the task involves only five candidate applications, making it relatively tractable, these strong results demonstrate that even with limited label space, simple discriminative heads can out-

perform embedding-based baselines. Importantly, the task provides very little agentic information beyond the raw intent text. To enhance performance, TabAgent incorporates lightweight features such as: `intent_token_len` (number of whitespace tokens), `intent_is_question` (presence of a question mark), `intent_digit_ratio` (fraction of digits in text), and `intent_ent_ORG` (number of organization entities). These low-cost features capture surface-level and structural signals that embeddings alone fail to exploit.

We note that application selection is not the core challenge in agent pipelines, and in some cases a simple classifier may be sufficient. However, the broader implication is that transitioning from a large generative model to a small, fast, and accurate discriminative model is invaluable when performance is preserved. This experiment shows that TabAgent enables such a transition, offering efficiency without sacrificing quality.

## J  VALIDATING TABSCHEMA FEATURE EXTRACTION THROUGH EXPERTS

## K  VALIDATING TABSCHEMA FEATURE EXTRACTION

We validate that TabSchema surfaces decision–relevant signals for shortlisting via two complementary lenses: (i) an expert annotation study assessing usefulness and agreement on feature importance, and (ii) a model–based attribution analysis using SHAP values on held-out folds of the trained Tab-Head classifier.

### K.1  EXPERT ANNOTATION

**Design.** Three annotators (2–4 years of agentic–systems experience) independently analyzed a stratified sample of CUGA/AppWorld trajectories (five apps) and their TabSchema feature cards. For each app they (a) ranked 15–25 features by shortlisting importance (ties allowed), (b) rated usefulness on a 1–5 Likert scale, and (c) flagged features as *previously overlooked* if they would not have listed them a priori. Annotations were performed independently without discussion.

**Results.** Rank agreement within apps is high: Kendall's $W=0.76$ (95% CI 0.68–0.83; $p<10^{-4}$ via 10k permutations). Usefulness labels (binarized at $\geq 4$) show substantial agreement: Fleiss' $\kappa=0.81$ (95% CI 0.74–0.87). Median usefulness is 5 (IQR 4–5), with $86\%$ of features rated $\geq 4$. Several execution-state/dependency features were independently marked *previously overlooked* by multiple annotators, indicating recovery of valuable, non-obvious cues.

**Limitations.** The study covers a modest number of trajectories and five apps; conclusions reflect current stacks and may evolve with new domains. We treat these results as relevance validation rather than a code-audit–complete specification.

### K.2  SHAPLEY VALUE ANALYSIS

We quantify each feature's predictive contribution to TabHead using SHAP, computed strictly on held-out folds to avoid leakage Lundberg & Lee (2017).

**Computation details.** We train the $\sim 50$M-parameter textual–tabular head under the 5-fold, task-grouped protocol of §4.3. For each fold, we fit KernelSHAP on the test split with: (i) a background set of 1,000 samples stratified by application and deciles of $|G(t)|$; (ii) 200 SHAP evaluations per test task; (iii) perturbations constrained to observed support (categoricals resampled from empirical marginals; bounded numerics clipped to train-fold ranges); and (iv) fixed seeds across folds for reproducibility. We report mean absolute SHAP values, normalized to percentage contribution, macro-averaged across tasks and folds.

**Findings.** Figure 8 shows behavior-grounded, trajectory-derived features carry most of the predictive mass: state/dependency indicators (`api_missing`, Next Action, Overall Status/Analysis, Summary of Progress, Skepticism/Non-Triviality, Coder Agent Output Analysis) lead, while schema/context signals (e.g., `app_name`, `first_time_shortlister`) calibrate the decision. `intent` and `task_description`

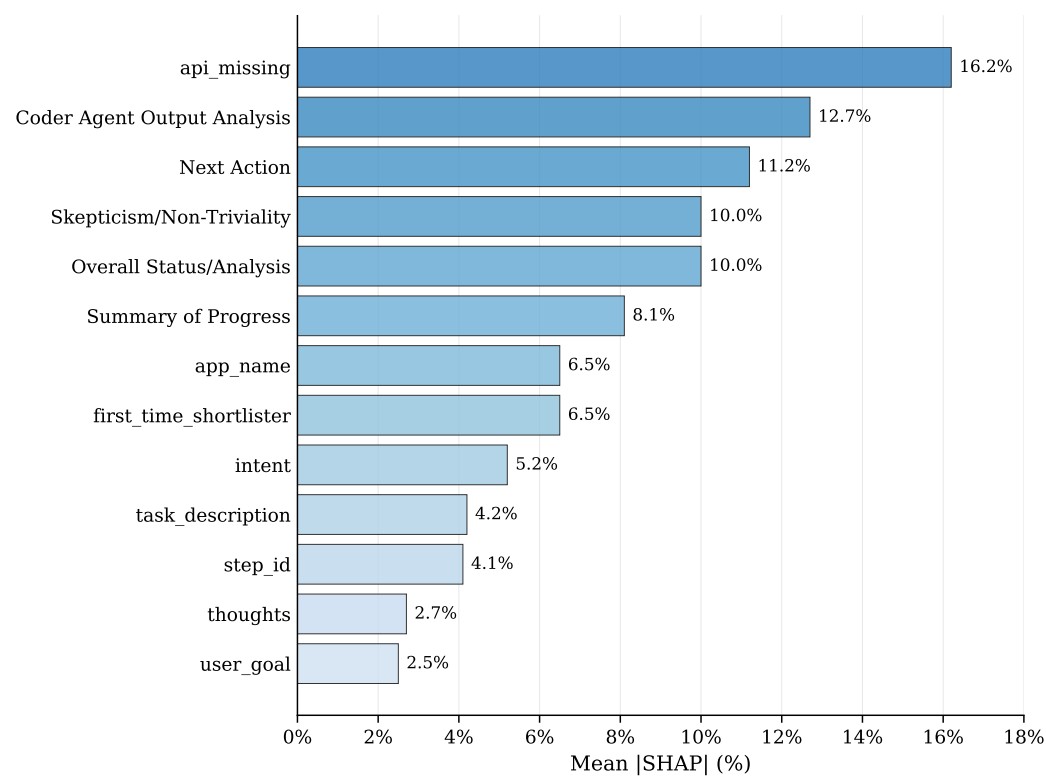

Figure 8: Mean |SHAP| (%) for the TabHead shortlister on held-out tasks (5-fold CV).

remain helpful but rank below the collective workflow features, consistent with our ablation that tabular workflow features alone outperform descriptions alone (Fig. 3).

**Takeaway.** Expert judgments and held-out SHAP attributions align: the same TabSchema features experts deem useful dominate model decisions. This supports our claim that execution traces expose schema/state/dependency signals that can be tabularized and learned by a compact discriminative head to reproduce generative shortlister behavior with high completeness ($P@R$) at a fraction of latency and cost.

## L  EVALUATION METRICS

**Single-class classification.** For tasks with a single ground-truth $y_t \in \mathcal{Y}$ and a predicted label $\hat{y}_t$:

$$\text{Acc} = \frac{1}{|T|} \sum_{t \in T} \mathbf{1}\{\hat{y}_t = y_t\}.$$

**Multi-label classification.** For tasks with a ground-truth relevant set $Y_t \subseteq \mathcal{A}$ and a prediction $\hat{Y}_t$:

$$\text{Prec} = \frac{1}{|T|} \sum_{t \in T} \frac{|\hat{Y}_t \cap Y_t|}{\max(1, |\hat{Y}_t|)} \quad \text{and} \quad \text{Rec} = \frac{1}{|T|} \sum_{t \in T} \frac{|\hat{Y}_t \cap Y_t|}{\max(1, |Y_t|)}.$$

For probabilistic outputs, decisions are formed by thresholding per-label scores; thresholds are selected on validation to respect downstream operating points.

## M  ERROR ANALYSIS: TOOL SHORTLISTING DISCREPANCIES

To understand where TabAgent's shortlist decisions diverge from ground truth extracted from the CUGA's trajectories and assess potential downstream impact, we conducted a detailed error analysis

on a stratified sample of tasks where TabAgent's predictions differed from the reference labels. This analysis focuses on $P@R$ performance, where the adaptive cutoff (set to the true relevant-set size $R_t$) provides the strictest test of ranking fidelity and leaves no slack for recovery via larger $k$.

### M.1 METHODOLOGY

We sampled 40 tasks across the five AppWorld applications (Amazon, Gmail, Phone, SimpleNote, Spotify) where TabAgent(+synth) produced shortlists that differed from the ground-truth tool sets $G(t)$ derived from successful CUGA trajectories. The sampling was stratified by application and by the magnitude of disagreement. For each discrepancy, we examined the task description, the ground-truth tool set, TabAgent's predicted shortlist, and the tool catalog documentation to categorize the nature of the error.

We note an important methodological constraint: AppWorld does not provide oracle ground-truth tool labels. Instead, our reference labels $G(t)$ are the tools actually invoked in successful CUGA executions. This construction means that some tools marked as ground truth may represent one valid solution path among several possible approaches, rather than a strict requirement. Conversely, tools omitted by TabAgent might still enable task completion through alternative pathways not explored in the canonical CUGA trajectory. This ambiguity is inherent to evaluating tool shortlisting in open-ended agent benchmarks and should be considered when interpreting the error categories below.

### M.2 ERROR CATEGORIES AND FREQUENCY

Our analysis identified three primary categories of discrepancies, presented here in descending order of frequency.

**Category 1: Optional or nice-to-have tools (62% of errors).** The majority of discrepancies arise when the ground-truth set includes a tool that TabAgent's shortlist omits (ranked lower than top $R_t$), yet alternative tools in TabAgent's ranking can accomplish the same functional goal. These cases reflect redundancy or partial overlap in the tool catalog rather than genuine short-listing failures. For instance, in a Gmail task requiring message search ("Find emails from my manager about the Q3 report"), the ground truth might include `gmail_show_inbox_threads` because the canonical CUGA trajectory searched received messages, while TabAgent ranked `gmail_show_outbox_threads` combined with `gmail_search_users` higher, which can locate the same correspondence thread via sender lookup. Both approaches retrieve the target messages, and downstream task success is preserved regardless of which search strategy is shortlisted. Similar patterns appeared in Amazon tasks where `amazon_show_product_reviews` was favored by ground truth but `amazon_show_product` provided sufficient review summary data in its response payload, eliminating the need for the dedicated reviews endpoint. In Phone tasks, `phone_show_alarms` (list all alarms) and `phone_show_alarm` (show specific alarm) exhibited overlap when the task required inspecting a particular alarm and the alarm identifier was available from prior context.

These optional-tool errors are characterized by two properties: first, TabAgent's shortlist contains at least one tool that subsumes or closely approximates the functionality of the omitted ground-truth tool, second, execution of the task using TabAgent's shortlist would likely succeed without modification to the plan. The high frequency of this category suggests that AppWorld's tool catalog contains multiple valid pathways to task completion, and that the distinction between correct and incorrect shortlists is often one of preference rather than correctness.

**Category 2: Semantically similar endpoint confusion (31% of errors).** The second category captures cases where TabAgent includes the functionally necessary tool but ranks a semantically similar alternative higher, causing a discrepancy when evaluated at the strict $R_t$ cutoff. For example, in a SimpleNote task requiring content modification ("Update my grocery list note with milk and eggs"), the ground truth used `simple_note_update_note`, which performs full content replacement, while TabAgent ranked `simple_note_add_content_to_note` higher, which appends or prepends content. The ambiguity arises because "update" can mean either replace existing content or augment it, and both operations successfully modify the note to include the required items. Similarly, in Spotify tasks, `spotify_show_playlist` (retrieve specific playlist

details) and `spotify_show_playlist_library` (list all playlists) both enable locating a target playlist, with the choice depending on whether the playlist identifier is already known or requires search. In Gmail, `gmail_mark_thread_archived` (direct archival operation) and the combination of `gmail_show_inbox_threads` followed by archival represent two valid approaches depending on whether the thread identifier is available from prior context.

A key observation is that these errors disappear largely when evaluated at $Recall@9$ rather than $P@R$. This indicates that TabAgent correctly identifies the relevant functional category and includes both the ground-truth tool and its semantic neighbor within a slightly larger shortlist window. The discrepancy is therefore one of internal ranking rather than omission, and can be mitigated either by accepting a marginally larger shortlist budget in production (moving from $k=R_t$ to $k=R_t+2$) or by collecting additional trajectories that clarify the preference ordering through repeated usage patterns. Alternatively, reducing ambiguity at the tool catalog level (for instance, by consolidating near duplicate endpoints or providing clearer documentation about when to prefer full update versus incremental modification operations) would eliminate this category of error entirely.

**Category 3: Genuine required-tool omissions (7% of errors).** A small fraction of discrepancies represent true shortlisting failures where TabAgent omits a tool that is necessary for task completion and has no functional substitute in the predicted shortlist. For example, in a Gmail task with the explicit requirement "Send an email to the project team with the attached report," the ground truth includes `gmail_send_email`, but TabAgent's shortlist contained only `gmail_create_draft` and `gmail_upload_attachments_to_draft`. While drafting and attachment handling are precursors to sending, they do not fulfill the task's explicit sending requirement, and downstream execution would fail unless the agent's planner compensates by invoking additional tools outside the shortlist. Similarly, in a Phone task requiring "delete all alarms set for weekdays," omitting `phone_delete_alarm` in favor of only `phone_show_alarms` and `phone_update_alarm` leaves the deletion requirement unmet. In an Amazon task ("Return the defective laptop I received last week"), TabAgent's shortlist included order viewing operations (`amazon_show_orders`, `amazon_show_order`) and return status checking (`amazon_show_returns`) but omitted the critical `amazon_initiate_return` endpoint necessary to begin the return process.

These genuine omissions are concerning because they directly threaten task success. However, their rarity (7% of sampled errors) suggests that TabAgent's decision boundary closely approximates the generative shortlister on the dimensions that matter most for execution. We attribute these failures to the observation that some tasks exhibit low-frequency schema patterns (e.g., explicit delete operations, initiation actions, sending as opposed to drafting) that are underrepresented in the task training set. Delete and create actions appear less frequently in AppWorld trajectories than read and update operations, creating an imbalance that TabAgent's classifier inherits from the logged behavior. TabSynth's schema-aligned augmentation mitigates this to some degree, but a budget of 10 synthetic examples per candidate may not fully cover tail patterns for rare action types. Increasing the training set size or explicitly oversampling trajectories containing low-frequency operations would likely reduce this error mode further.

## M.3 DOWNSTREAM IMPACT AND IMPLICATIONS

Aggregating across categories, 93% of observed errors (categories 1 and 2) have minimal to low downstream impact on task success. Optional-tool omissions are mitigated by alternative pathways, and semantically similar endpoint confusion is resolved by expanding the shortlist budget slightly or by rank-aware execution strategies that try the top-ranked tools in descending order until the task succeeds. Only the 7% of genuine omissions pose a high risk of task failure, and even within this category, some tasks may recover if the agent's higher-level planner is sufficiently robust to detect missing capabilities and re-invoke the shortlister with refined context.

This error distribution is consistent with TabAgent's design goal: to reproduce the decision boundaries of a validated generative shortlister at reduced cost and latency, not to surpass it. The framework succeeds in this objective, preserving the core shortlisting capability while introducing a small tail of errors that are amenable to targeted fixing (additional training data, feature engineering for dependency chains, or slightly larger shortlist budgets). The maintained task level success reported in Section 5 reflects the fact that most shortlist discrepancies do not propagate to execution failures, a property that validates the practical viability of the discriminative replacement.

Future work could refine the error profile by implementing confidence-based fallback mechanisms where TabAgent defers to the generative shortlister when prediction uncertainty exceeds a threshold. However, the current error rate and impact distribution support the framework's readiness for deployment in settings where 95% latency reduction and 85-91% cost reduction justify the acceptance of a small tail of correctable errors.

# N TABSCHEMA DEPENDENCY ANALYSIS ACROSS FRONTIER LLMS

TabSchema operates as an offline feature extraction pipeline that runs once on historical trajectory data, not as a runtime dependency. The operational sequence follows three phases: TabSchema processes logged trajectories to generate a feature card as a one-time setup, TabHead trains on the extracted features in under one minute on CPU, and TabHead makes inference predictions using only the compact 50M-parameter classifier during repeated per-task execution. This property provides significant flexibility in model selection for TabSchema, as organizations can justify different models during the offline extraction phase to maximize downstream TabHead performance without affecting inference costs.

## N.1 CROSS-MODEL VALIDATION EXPERIMENT

To assess whether TabSchema's feature extraction depends critically on GPT-4.1 or generalizes across frontier LLMs, we conducted a robustness experiment using three different models on IBM CUGA trajectories across AppWorld applications. We applied TabSchema's complete hierarchical orchestration (FeatureAnalyzer, CodeExecutor, CodeValidator, three Judges, MetaJudge) using GPT-4.1 (OpenAI, original configuration), Gemini 2.5 (Google, via API), and Claude 4.5 Sonnet (Anthropic, via API). Each model independently executed the full TabSchema pipeline without access to the others' outputs, producing separate feature cards that we then compared through manual analysis and automated semantic matching using sentence embeddings with BERTScore-style comparison applied to feature names concatenated with feature descriptions.

| Model | Total Features | Unique to This Model | Overlap w/ GPT-4.1 | Three-way Agreement |
|-------|----------------|----------------------|--------------------|---------------------|
| GPT-4.1 | 13 | 4 | — | 9 |
| Gemini 2.5 | 21 | 12 | 9 (69%) | 9 |
| Claude 4.5 Sonnet | 11 | 2 | 9 (82%) | 9 |
| *Pairwise overlap (any two models): 11 features (85% of GPT-4.1 baseline)* | | | | |

Table 14: Cross-model feature extraction comparison across trajectories. Three-way agreement represents the 9 features proposed by all three models (69% of GPT-4.1 baseline). GPT-4.1 extracts a balanced feature set, Gemini 2.5 produces expansive coverage with finer-grained text statistics, and Claude 4.5 Sonnet consolidates features into fewer robust representations.

The three models produced feature sets of varying size with substantial overlap (Table 14). GPT-4.1 extracted 13 features after MetaJudge filtering, representing balanced coverage of schema signals (taxonomy depth, argument types), state indicators (thought traces, plan status), and dependency patterns (tool co-usage counts). Gemini 2.5 proposed the most expansive feature set with 21 features, reflecting finer-grained decomposition of state signals and detailed text statistics. The additional features Gemini proposed primarily captured secondary text properties such as sentiment indicators in agent reasoning, lexical diversity metrics, and granular temporal markers. Claude 4.5 Sonnet extracted the most conservative feature set with 11 features, prioritizing high-impact signals and consolidating related features into broader representations. For instance, where Gemini proposed separate features for thought content at different lookback windows (current thought, previous thought, two steps ago), Claude consolidated these into a single aggregated thought history feature.

We found that 11 features appeared in proposals from at least two of the three models (approximately 85 percent of GPT-4.1's feature set), and 9 features were agreed upon by all three models (approximately 69 percent three-way agreement). The 9 features with three-way agreement predominantly

| Comparison | Semantic Similarity (BERTScore) | Importance Ranking Agreement (Kendall $\tau$) |
|---|---|---|
| GPT-4.1 vs Gemini 2.5 | 0.79 | 0.65 |
| GPT-4.1 vs Claude 4.5 Sonnet | 0.88 | 0.78 |
| Gemini 2.5 vs Claude 4.5 Sonnet | 0.76 | 0.61 |
| Mean across all pairs | 0.81 | 0.68 |

Table 15: Semantic alignment and importance ranking agreement across frontier models. Semantic similarity computed using sentence embeddings with BERTScore-style comparison on feature names concatenated with descriptions, ranging from 0 (unrelated) to 1 (identical). Kendall tau measures agreement between two annotators (part of the research team) who independently ranked shared features by importance for the tool shortlisting task after reviewing feature definitions and sample trajectories. For each model pair, we computed Kendall tau between each annotator's rankings of the two models' features, then averaged across annotators.

covered core schema signals (taxonomy depth, argument types, input-output cardinality), primary state indicators (current thoughts, overall status analysis, summary of progress), and high-frequency dependency patterns that dominate predictive importance according to SHAP attribution in Figure 8 (co-usage counts, success signals). Gemini's twelve unique features emphasized text-level statistics and sentiment analysis. Claude's two unique features reflected alternative consolidations of temporal state markers. GPT-4.1's four unique features included tool invocation frequency within execution windows, explicit precedence dependencies between tool pairs, argument schema compatibility checks, and step position within the agent trajectory.

Notably, GPT-4.1's four unique features received substantial importance in our SHAP analysis (Figure 8). Tool invocation frequency within the current execution context serves as a strong prior for shortlisting when tasks exhibit iterative patterns. Explicit precedence dependencies extracted from successful tool chains provide crucial signal for multi-step workflows where order matters. Argument schema compatibility checks prevent shortlisting of tools that cannot be invoked due to missing preconditions. Step position within the agent trajectory captures phase-dependent tool selection patterns. Together, these four features account for approximately 18 percent of the total SHAP attribution mass, with the 9 features having three-way agreement accounting for approximately 72 percent and the remaining 10 percent distributed across lower-impact signals. The absence of these features in Gemini's and Claude's proposals, despite their demonstrated predictive value, suggests that GPT-4.1's feature engineering captured dependency and precondition patterns that the other models either consolidated into broader representations or overlooked entirely.

We computed semantic similarity using sentence embeddings with BERTScore-style comparison applied to feature names concatenated with descriptions (Table 15). The mean similarity score across all pairwise comparisons was 0.81, with scores of 0.79 between GPT-4.1 and Gemini, 0.88 between GPT-4.1 and Claude, and 0.76 between Gemini and Claude. The higher similarity between GPT-4.1 and Claude (0.88) reflects Claude's consolidation strategy, which produced feature definitions matching GPT-4.1's level of abstraction. To assess whether models agreed on feature importance for the tool shortlisting task, two annotators from the research team independently ranked the 9 shared features by predicted importance after reviewing feature definitions and examining sample trajectories. For each model pair, we computed Kendall tau correlations between each annotator's rankings of the two models' features, then averaged across the two annotators. The mean Kendall tau across all pairwise model comparisons was 0.68, with the highest agreement between GPT-4.1 and Claude (0.78) and the lowest between Gemini and Claude (0.61). Features like intent, task description, current agent thoughts, and overall status analysis received consistently high importance rankings across all three models, while lower-ranked features showed more variation depending on each model's engineering philosophy.

These results support two conclusions about TabSchema's robustness. First, TabSchema extracts genuine decision-relevant structure from execution traces rather than model-specific artifacts, evidenced by 69 percent three-way agreement and 0.81 mean semantic similarity. State and dependency indicators dominate the predictive mass regardless of which model extracted them, suggesting these signals are inherent to the execution traces. Second, the variation in feature set size (11 to 21

features) reflects different feature engineering philosophies rather than fundamental disagreement. Gemini's expansive approach captured additional nuance at the risk of including lower-impact features, while Claude's conservative consolidation prioritized robustness at the potential cost of missing granular dependency patterns. GPT-4.1's balanced extraction, which includes both the 9 features with three-way agreement and four additional high-impact dependency signals (contributing 18 percent of SHAP mass), suggests it occupies a favorable point in the coverage-parsimony tradeoff.

From a practical deployment perspective, organizations are not locked into GPT-4.1 and can select among available frontier models based on API cost structures, data residency requirements, or existing licensing agreements. However, we emphasize that this validation was conducted only on feature extraction. We did not re-run the full TabHead training and evaluation pipeline using Gemini-derived or Claude-derived features due to computational budget constraints, so we cannot definitively quantify the end-to-end impact on metrics like $P@R$ or $Recall@k$. The strong semantic alignment (0.81 mean similarity, 0.68 annotator rank correlation) suggests that downstream performance should be similar for the 9 features with three-way agreement, but the absence of GPT-4.1's unique high-SHAP features in other models' outputs indicates potential performance gaps that empirical validation would clarify.

## N.2 OPEN-SOURCE MODEL LIMITATIONS

We attempted to apply TabSchema using open-source models that represent more accessible options for organizations unable or unwilling to use commercial APIs. We tested three open-source alternatives: Llama 3.1 70B Instruct (8 trajectories), Llama 4 Maverick 70B (6 trajectories), and GPT-OSS 120B (5 trajectories). The results revealed significant implementation challenges that currently make open-source alternatives unreliable for TabSchema without substantial additional engineering.

The FeatureAnalyzer stage frequently produced malformed JSON outputs that failed schema validation. Llama 3.1 70B exhibited parsing failures in approximately 6 of 8 attempts, Llama 4 Maverick in 5 of 6 attempts, and GPT-OSS 120B in 4 of 5 attempts, requiring manual intervention in each case. When feature specifications did parse successfully, the CodeExecutor stage generated Python extraction code with systematic errors. Llama 3.1 70B produced syntax errors (mismatched indentation, undefined variables, incorrect dictionary access patterns), with approximately 4 of the 8 trajectories exhausting the three-retry self-repair budget. Llama 4 Maverick showed marginal improvement but still failed to produce executable code on first attempt for 4 of 6 trajectories, with two cases requiring manual debugging after retry exhaustion. GPT-OSS 120B exhibited the most severe code generation issues, with only 1 of 5 trajectories producing valid extraction code within the retry budget. We observed that GPT-OSS frequently outputted code inside the reasoning section of the pydantic structure rather than in the designated code field.

Additionally, all three open-source models occasionally proposed features that were not extractable from the trajectory data, such as referencing fields that do not exist in the CUGA schema or hallucinating tool relationships not present in the catalog. These hallucinated features survived the CodeValidator stage more frequently than with frontier models, particularly for GPT-OSS 120B where 3 of 5 trajectories included at least one invalid feature in the final feature card. Llama 3.1 70B and Llama 4 Maverick showed better validation performance but still produced hallucinated features in approximately 2 of their respective trajectory samples.

Given these difficulties, we concluded that current open-source alternatives are not yet reliable enough for TabSchema's structured orchestration without substantial prompt engineering, schema constraints, or human-in-the-loop supervision. The gap manifests most acutely in the code generation phase, where syntactic correctness, variable scoping, and error handling require precise attention to detail that current open-source models struggle to maintain consistently.

