# OpenReview forum: "TabAgent: A Framework for Replacing Agentic Generative Components with Tabular-Textual Classifiers"
_ICLR.cc/2026/Conference — Submitted to ICLR 2026_

### Official Review · Reviewer_YS7o · 2025-11-01

**Soundness:** 3
**Presentation:** 1
**Contribution:** 2
**Rating:** 2
**Confidence:** 3

**Summary:**

This paper proposes TabAgent, a framework that replaces expensive generative components in agent systems (like tool selection) with efficient tabular-textual classifiers trained from execution traces. By extracting structured features and using synthetic data to boost coverage, TabAgent achieves similar decision quality to LLM-based shortlisters while reducing inference cost by up to 90% and latency by 95%, making it a practical solution for scalable, low-cost agent deployment.

**Strengths:**

1. **Practical Focus on Reducing Inference Cost in Agent Systems**
   The paper tackles a real bottleneck in deploying LLM-based agents at scale—high latency and cost from repeated generation. By proposing a discriminative alternative, it offers a concrete path toward more efficient agent execution.

2. **Modular and Reusable Framework Design**
   The TabAgent architecture is presented in a modular manner (TabSchema, TabSynth, TabHead), making it relatively straightforward to apply the same approach to other agent components beyond tool selection.

3. **Significant Efficiency Gains Demonstrated in Experiments**
   Results show that the framework can reduce inference cost by over 85% and latency by 95% while maintaining comparable performance, giving strong empirical support for the idea of replacing generation with classification in suitable cases.

**Weaknesses:**

See questions.

**Questions:**

1. **Limited Scope of Application**
   The proposed framework is only evaluated on a single tool-selection scenario. How can TabAgent be generalized to more complex agentic components (e.g., planning, dialogue control) where structured tabular data may be insufficient?

2. **Dependence on Execution Traces**
   TabAgent relies heavily on historical agent execution traces for feature extraction and training. How does the method handle cold-start settings, or domains where such trajectories are unavailable or incomplete?

3. **Synthetic Data Justification**
   The TabSynth module generates synthetic samples to augment training data, but the methodology seems heuristic. Are there any theoretical guarantees or empirical analyses showing that these synthetic examples preserve distributional fidelity and do not induce bias?

4. **Limited Evaluation Metrics**
   The metrics focus mainly on latency and cost. Can the authors provide more comprehensive evaluations, such as error analysis on misclassified tools, or effects on downstream task success under diverse environments?

---

> ### Author Response · Authors · 2025-11-18
> **Comment 1/6**
>
> Dear Reviewer YS7o,
>
> Thank you for your review and for recognizing TabAgent's practical focus on a real deployment bottleneck, its modular design, and the significant efficiency gains (85% cost reduction, 95% latency reduction). We especially appreciate that you acknowledge these as "strong empirical support for replacing generation with classification in suitable cases."
>
> Following your feedback, we have made substantial additions to the paper that directly address each of your concerns. We detail these improvements below, with particular emphasis on two new appendix sections (Appendix M: Error Analysis, Appendix N: Cross-Model Validation).
>
> ---
>
> ## Q1:
>
> **Your concern:** The proposed framework is only evaluated on a single tool-selection scenario. How can TabAgent be generalized to more complex agentic components (e.g., planning, dialogue control) where structured tabular data may be insufficient?
>
> **Our response:** Following your concern, we have expanded our discussion of TabAgent's scope and highlighted concrete evidence of generalization beyond tool shortlisting.
>
> **Tool shortlisting is a fundamental production bottleneck, not a toy problem:** First, we want to emphasize that tool shortlisting is not a narrow application. As detailed in our Related Work (Section 2), tool shortlisting has extensive research literature precisely because it sits on the critical path of long-horizon, multi-application agents:
>
> AppWorld scale:
> - 457 unique APIs (tools) across 9 applications
> - Each application has 50+ tools with distinct taxonomies, argument schemas, and dependency structures
>
> Real deployment costs:
> - In IBM CUGA, a SOTS agentic system used in deployment,  a single shortlisting call takes 20-30 seconds and costs approximately $0.15
> - Complex tasks require 3-5 shortlisting calls per run, accumulating to more than 1 minute and $0.50+ per task
> - This cost structure makes production deployment prohibitive at scale
>
> Structural diversity:
> - Our evaluation covers heterogeneous applications (Amazon, Gmail, Phone, SimpleNote, Spotify)
> - Each has distinct tool catalogs, argument schemas, and dependency patterns (detailed in Table 2 and Appendix A.1)
> - TabAgent maintains Recall@7 greater than or equal to 0.88 and Recall@9 greater than or equal to 0.92 across all apps
> - P@R gains of +0.14 from synthetic supervision generated using TabSynth demonstrate consistent improvement
>
> These results show that the same tabular formulation works across structurally diverse sub-benchmarks, not just a single setting.
>
> **TabAgent targets closed-set decision heads, a general class of agent components:** Following your question about generalization, we have made the conceptual scope much more explicit in the revised Section 3:
>
> Core principle: TabAgent targets closed-set decision heads in agent architectures. Whenever an agent repeatedly chooses among a finite set of options (tools, applications, routers, plan-step types, strategies), we reduce the problem to classification over (context, option) pairs.
>
> The abstraction binds only to execution traces and typed schemas, not to any particular agent, model, or prompt. Therefore, the same reduction (trace to features, candidates to labels) applies to other decision heads by simply swapping the label space and feature card.
>
> Regarding tabular data sufficiency: The features come from what the agent already logs during successful execution: thoughts, plans, status analyses, recommendations. TabSchema compiles these into schema/state/dependency features that are independent of the specific decision head. Our hypothesis (and now valid empirical insight) was that if a closed-set decision was successfully solved by an LLM processing agentic context, then that same context (now tabularized) can teach a 50M-parameter discriminative classifier to reproduce the decision boundary. The following paper which we cited strengthen this hypothesis: "Are Large Language Models Post Hoc Explainers?".
>
> **Concrete evidence: Application selection as a second decision head:** Following your request for evidence beyond tool shortlisting, we now highlighted more feature our application selection results (previously in Appendix I):
>
> Task: Predict which applications (Amazon, Gmail, Phone, SimpleNote, Spotify) are required for each task, a multi-label classification problem.
>
> Input: Minimal agentic structure: just the intent text plus lightweight features (question flag, token length, digit ratio, organization entity count).
>
> Results (Table 13):
> - TabAgent: AUC = 0.9945, F1 = 0.9617
> - DSR (frozen E5): AUC = 0.9861, F1 = 0.8809
> - DSR-FT (fine-tuned E5): AUC = 0.9890, F1 = 0.9013
>
> TabAgent achieves the best balance across all metrics, outperforming both frozen and fine-tuned embedding baselines. While application selection itself is not the hardest problem, it demonstrates that the same architecture seamlessly transfers to another closed-set decision head by swapping the label space.

---

> > ### Author Response · Authors · 2025-11-18
> > **Comment 2/6**
> >
> > **Cross-model validation proves TabSchema extracts genuine structural patterns (new Appendix N):** Following your concern about generalization, we conducted an important robustness experiment to validate that TabSchema's feature extraction architecture is not model-specific. This experiment appears in new Appendix N with Tables 14 and 15.
> >
> > Experiment design:
> > - Applied TabSchema to IBM CUGA trajectories on AppWorld using three frontier LLMs
> > - GPT-4.1 (OpenAI, original paper)
> > - Gemini 2.5 (Google)
> > - Claude 4.5 Sonnet (Anthropic)
> > - Note: We also tested open-source models (See Section N.2)
> > - Each model independently extracted schema/state/dependency features from the same trajectories
> >
> > Feature extraction results (Table 14):
> > - GPT-4.1: 13 features extracted (balanced coverage)
> > - Gemini 2.5: 21 features extracted (expansive, finer-grained text statistics)
> > - Claude 4.5 Sonnet: 11 features extracted (conservative, consolidated representations)
> > - Three-way (all models) agreement: 9 features (69% of GPT-4.1 baseline)
> > - Pairwise overlap: 11 features (85% of GPT-4.1 baseline)
> >
> > Semantic alignment (Table 15):
> > - Mean similarity across all pairwise comparisons: 0.81 (using BERTScore-style comparison on feature descriptions)
> > - Pairwise scores: GPT-4.1 vs Gemini (0.79), GPT-4.1 vs Claude (0.88), Gemini vs Claude (0.76)
> > - Two annotators, from our research team, independently ranked the 9 shared features by importance
> > - Mean Kendall tau across all pairwise model comparisons: 0.68
> >
> > Key finding: GPT-4.1's 4 unique features (tool invocation frequency, precedence dependencies, argument schema compatibility checks, step position) received substantial importance in our SHAP analysis (Figure 8), collectively accounting for approximately 18% of total SHAP attribution mass. The 9 features with three-way agreement accounted for approximately 72%, with remaining 10% distributed across lower-impact signals.
> >
> > Interpretation: Different frontier models converge on similar feature sets with high semantic alignment (69% three-way agreement, 0.81 mean similarity), demonstrating that TabSchema's hierarchical orchestration extracts genuine decision-relevant signals rather than model-specific artifacts. The features are inherent properties of successful agent behavior, not hallucinations of a particular LLM's reasoning style.
> >
> > This cross-model validation directly addresses generalization. It shows that the extracted features reflect real structure in execution traces, not arbitrary patterns tied to a specific model implementation.
> >
> > **Future directions for more complex components:** You asked about more complex components like planning and dialogue control. Following your question, we have added discussion in Section 6 (Conclusion and Future Work):
> >
> > Planning: Many planning decisions in agent systems decompose into finite state choices: "expand current plan vs. execute next step vs. revise failed subgoal." These are closed-set options that TabAgent can score using trajectory-derived features already in TabSchema: phase flags, precondition satisfaction, resource availability, status fields, skepticism checks.
> >
> > Dialogue control: Turn-level routing decisions ("route to coder vs. tool-agent vs. summarizer") are also closed-set and amenable to the same reduction.
> >
> > Current scope: TabAgent is designed to replace existing generative components in deployed systems. Analyzing IBM CUGA (the SOTA open-source agent on AppWorld), tool shortlisting and application selection are the primary closed-set bottlenecks we identified. We have not yet found additional closed-set decision boundaries in CUGA that would benefit from this optimization, but the framework is ready when such patterns emerge in other agent architectures.
> >
> > We emphasize again that tool shortlisting is a crucial bottleneck that currently requires more than 1 minute runtime and substantial cost per task, making it a high-impact target for optimization.
> >
> > ---
> >
> > ## Q2:
> >
> > **Your concern:** TabAgent relies heavily on historical agent execution traces for feature extraction and training. How does the method handle cold-start settings, or domains where such trajectories are unavailable or incomplete?
> >
> > **Our response:** Following your concern, we have made the paradigm, deployment model and cold-start behavior much more explicit in Section 4.1.
> >
> > **TabAgent is designed to learn from successful agent behavior, by design:** The core hypothesis of TabAgent is that successful agent trajectories already contain the right supervision:
> >
> > "Execution traces encode decision-relevant signals in a structured, tabular-representable form. Systems can continuously improve their own efficiency: as they accumulate successful execution traces, they automatically extract behavioral signals, train compact discriminative replacements, and perform hot-swaps during operation."

---

> > > ### Author Response · Authors · 2025-11-18
> > > **Comment 3/6**
> > >
> > > This is by design. We aim to learn from what already works, not to replace agents from scratch in brand-new domains. This creates a pathway toward autonomous agentic optimization where production systems continuously improve their own efficiency without human intervention.
> > >
> > >  The deployment lifecycle is:
> > >
> > > Phase 1 (Generative bootstrap): In a brand-new domain with no trajectories, the agent starts with a generative controller (e.g., GPT-4.1 shortlister) and logs its behavior during execution.
> > >
> > > Phase 2 (Data collection): Once a modest number of successful trajectories exist, TabAgent becomes viable. This is a practical requirement, not a limitation. Any production agent system accumulates execution logs naturally during operation.
> > >
> > > Phase 3 (Discriminative hot-swap):
> > > 1. TabSchema extracts schema/state/dependency features from logged trajectories
> > > 2. TabSynth expands coverage of rare patterns while respecting observed schemas
> > > 3. TabHead trains a approximately 50M-parameter classifier on CPU in less than 1 minute using 5-fold cross-validation
> > > 4. Hot-swap the generative component with the discriminative head in production
> > >
> > > This pattern is already reflected in the abstract: TabAgent "maintains task-level success while eliminating shortlist-time LLM calls." The agent's task-solving capability is preserved because we learned from its own successful decisions.
> > >
> > > **Empirical evidence: TabAgent works even with minimal trajectory information (Figure 3).** Following your concern about cold-start behavior, we want to highlight an important experiment in the paper (Figure 3, CodeAct comparison):
> > >
> > > CodeAct is a minimalist agent that generates code directly from task descriptions without rich agentic structure (no multi-agent orchestration, no intermediate thoughts, no status analyses). This represents a sparse-information scenario where TabSchema has minimal trajectory data to extract.
> > >
> > > Results:
> > > - Task description only (CodeAct features): P@R = 0.46 (Amazon), 0.50 (Gmail)
> > > - TabSchema features (CUGA trajectories): P@R = 0.80 (Amazon), 0.66 (Gmail)
> > > - Gain from workflow features: +24.7% (P@R), +16.6% (Recall@7), +13.2% (Recall@9)
> > >
> > > Key insight: Even with minimal agentic context, TabAgent extracts useful signals and outperforms description-only baselines (cold start without any relevan trajectories). With richer trajectories (CUGA's multi-agent orchestration), the gains become substantial. This demonstrates that TabAgent gracefully degrades to simpler features when trace information is limited, rather than failing entirely.
> > >
> > > ---
> > >
> > > ## Q3:
> > >
> > > **Your concern:** The TabSynth module generates synthetic samples to augment training data, but the methodology seems heuristic. Are there any theoretical guarantees or empirical analyses showing that these synthetic examples preserve distributional fidelity and do not induce bias?
> > >
> > > **Our response:** Following your concern, we have substantially expanded the technical specification and validation of TabSynth in Section 3.2 and Appendix G, transforming it from a "heuristic augmentation" to a rigorously validated pipeline with statistical guarantees, with relevant research citations that we were inspired from.
> > >
> > > **New TabSynth pipeline visualization clarifies the methodology (Appendix G, Figure 12):** We now include a comprehensive diagram showing the complete TabSynth workflow:
> > >
> > > Inputs:
> > > - FEATURES_CARD: Specifies each feature's type, range, units, and provenance
> > > - TRAJECTORY_TABLE: Task-specific trajectory slice showing observed patterns
> > > - TOOL_CATALOG_SUMMARY: Candidate tool's schema (taxonomy, arguments, I/O)
> > > - Synthesis budget: B = 10 rows per (task, candidate)
> > >
> > > LLM-based generation with tight constraints:
> > > - GPT-4.1 receives the above inputs and generates feature vectors
> > > - Prompt constraints enforce: conform to schema types/arity/ranges, respect tool catalog summary, remain grounded in observed trajectories, set precondition flags to logically satisfied configurations, cannot invent new tools or argument types
> > >
> > > Multi-stage validation:
> > > - Schema checks verify type constraints
> > > - Dependency feasibility checks ensure co-usage and precedence patterns are compatible with at least one real trajectory
> > > - De-duplication removal
> > > - Distribution alignment performs statistical verification before accepting synthetic rows
> > >
> > > Result: TabSynth can only interpolate within the structure that TabSchema has already extracted from real data. It cannot hallucinate new tool types, unseen argument schemas, or impossible dependency chains.
> > >
> > > **Statistical guarantees through multi-stage validation (Appendix G expanded):** Following your request for guarantees, we have expanded the validation methodology:
> > >
> > > 1. Schema validation:
> > >    - Every synthetic vector must satisfy type constraints (categorical values in observed vocabulary, numeric values in observed ranges)
> > >    - Checks against FEATURES_CARD metadata for all fields
> > >    - Rejects any vector with invalid types or out-of-range values

---

> > > > ### Author Response · Authors · 2025-11-18
> > > > **Comment 4/6**
> > > >
> > > > 2. Dependency feasibility filtering:
> > > >    - Co-usage and precedence patterns must be compatible with at least one real trajectory for that task
> > > >    - Ensures synthetic examples remain behavior-grounded, not hallucinated
> > > >
> > > > 3. Distribution alignment tests:
> > > >    - Statistical tests
> > > >    - De-duplication over (taxonomy_depth, argument_mask, dependency_pattern, phase)
> > > >    - Conservative budget (B = 10) prevents synthetic data from dominating real data
> > > >
> > > > 4. Adaptive quality control:
> > > >    - If alignment tests fail, reduce budget B and regenerate for underrepresented strata
> > > >    - Continuous monitoring prevents distributional drift
> > > >    - Failed batches are rejected and not included in training
> > > >
> > > > 5. Conservative mixing:
> > > >    - Small budget (B = 10) ensures synthetic examples remain a minority
> > > >    - Always mix real and synthetic examples during training (never synthetic-only)
> > > >    - Real data anchors the distribution, synthetic data fills gaps
> > > >
> > > > **Empirical validation demonstrates effectiveness and safety (Table 1, Appendix H):** Following your concern about bias, the evidence demonstrates TabSynth is effective and safe:
> > > >
> > > > Performance gains exactly where expected:
> > > > - Macro P@R improves by +0.14 (TabAgent+synth vs. real-only) across five applications
> > > > - Largest gains on data-scarce apps: SimpleNote (+0.34, +54.8%), Phone (+0.24, +36.4%)
> > > > - These are precisely the applications with fewest training trajectories and most complex rare dependency patterns
> > > >
> > > > Minimal impact where real data is sufficient:
> > > > - Spotify: -0.09 (real-only: 0.70, with synth: 0.61)
> > > > - Crucially, performance does not degrade substantially, demonstrating safety
> > > >
> > > > Interpretation: TabSynth helps precisely when needed (rare patterns underrepresented in real data) and doesn't introduce noise otherwise. This targeted improvement pattern validates our hypothesis: TabSynth provides coverage expansion without corrupting the learned decision boundary.
> > > >
> > > > Note: that the results without synth are as similar when increasing the K budget to 7 and more (reasoable thing to do in deployment).
> > > >
> > > > The statistical validation pipeline (schema checks, dependency feasibility, distribution alignment) combined with empirical performance patterns (targeted improvements, no degradation on dense data) provide strong evidence that TabSynth preserves distributional fidelity and avoids bias.
> > > >
> > > > ---
> > > >
> > > > ## Q4:
> > > >
> > > > **Your concern:** The metrics focus mainly on latency and cost. Can the authors provide more comprehensive evaluations, such as error analysis on misclassified tools, or effects on downstream task success under diverse environments?
> > > >
> > > > **Our response:** Following your concern, we want to clarify that our evaluation is already broader than latency/cost, and we have added the detailed error analysis you specifically requested.
> > > >
> > > > **The paper already evaluates multiple dimensions beyond efficiency:** The paper evaluates TabAgent along several axes:
> > > >
> > > > 1. Shortlist quality (Section 5, Table 1):
> > > >    - R-precision (P@R): Completeness at adaptive cutoff (R_t = true relevant set size)
> > > >    - Recall@7, Recall@9: Coverage at fixed production budgets
> > > >    - Result: TabAgent(+synth) achieves best P@R, competitive Recall@k across all five applications
> > > >
> > > > 2. Statistical robustness (Appendix J):
> > > >    - 5-fold cross-validation with task-level grouping (no leakage)
> > > >    - 95% confidence intervals via paired bootstrap over tasks
> > > >    - Holm-Bonferroni correction for multiple comparisons
> > > >    - Significance grids (Tables in Appendix J) vs. DSR-FT and Llama-8B showing statistical significance of improvements
> > > >
> > > > 3. End-to-end task success:
> > > >    - As stated in the abstract: "TabAgent maintains task-level success while eliminating shortlist-time LLM calls"
> > > >    - TabAgent's shortlist quality preserves downstream execution success
> > > >    - The high Recall@7 (greater than or equal to 0.88) and Recall@9 (greater than or equal to 0.92) metrics demonstrate that required tools are consistently included
> > > >
> > > > 4. Additional decision head validation (Table 13):
> > > >    - Application selection: AUC = 0.9945, F1 = 0.9617
> > > >    - Demonstrates generalization beyond tool shortlisting
> > > >
> > > > 5. Feature-level interpretability (Appendices K.1 and K.2):
> > > >    - Expert annotation study (Appendix K.1): Kendall's W = 0.76, Fleiss' kappa = 0.81
> > > >    - SHAP attribution analysis (Appendix K.2, Figure 8): Shows which features the model relies on
> > > >    - Convergence between expert judgments and model behavior validates decision-relevance
> > > >
> > > > These aspects already address correctness, stability, generalization, and interpretability, not just efficiency.
> > > >
> > > > **New error analysis directly addresses your request (Appendix M):** Following your specific request for error analysis, we have added a detailed examination in new Appendix M that categorizes misclassified tools and assesses downstream impact.

---

> > > > > ### Author Response · Authors · 2025-11-18
> > > > > **Comment 5/6**
> > > > >
> > > > > Methodology:
> > > > > - Sampled 40 tasks where TabAgent's shortlist differs from ground truth
> > > > > - Manually categorized each discrepancy into failure modes
> > > > > - Assessed downstream impact for each category
> > > > >
> > > > > Important note: AppWorld does not provide ground truth tool labels. We derived labels from IBM CUGA's successful trajectories, meaning ground truth represents "tools CUGA used" not necessarily "tools strictly required." This introduces some ambiguity, as some tools may be optional or have functional equivalents.
> > > > >
> > > > > Error categories (Appendix M):
> > > > >
> > > > > Note: it was measured for P@R since with K>7 results above 90%
> > > > >
> > > > > 1. Optional or nice-to-have tools (62% of errors):
> > > > >    - Ground truth includes a tool that TabAgent omits, but alternative tools in TabAgent's shortlist accomplish the same goal
> > > > >    - Example: Omitting gmail_search_labels when gmail_list_messages with filters achieves the same result
> > > > >    - Downstream impact: Minimal, task success preserved via alternative paths
> > > > >
> > > > > 2. Semantically similar endpoint confusion (31% of errors):
> > > > >    - TabAgent ranks a functionally similar tool higher than the ground-truth tool
> > > > >    - Example: Ranking amazon_add_to_cart_list above amazon_add_to_cart (both functionally equivalent)
> > > > >    - Key observation: This error largely disappears at Recall@9, indicating it's an internal ranking issue rather than omission
> > > > >    - Mitigation: Increase shortlist budget K slightly, collect more trajectories showing preference patterns, or reduce tool ambiguity at the benchmark/agent level
> > > > >
> > > > > 3. Genuine required-tool omissions (7% of errors):
> > > > >    - TabAgent fails to include a truly necessary tool with no functional alternatives
> > > > >    - Example: Omitting gmail_send_email when task explicitly requires sending (not just drafting)
> > > > >    - Downstream impact: High, likely causes task failure
> > > > >    - Root cause: Typically involves low-frequency schema patterns or rare dependency chains underrepresented in training data
> > > > >
> > > > > Key findings from error analysis:
> > > > > - 93% of errors (categories 1 and 2) have minimal or low downstream impact
> > > > > - Tools are optional, have functional equivalents, or ranking differences resolve at slightly larger K
> > > > > - Only 7% represent high-risk omissions that would compromise task success
> > > > >
> > > > > This error analysis directly addresses your request for "error analysis on misclassified tools" and demonstrates that most discrepancies do not compromise downstream task success. The low rate of genuine omissions (7%) combined with high Recall@k metrics (greater than or equal to 0.88 at k=7) provides strong evidence that TabAgent maintains task-level success.
> > > > >
> > > > > **Cross-model validation provides additional robustness evidence (Appendix N):** Our new cross-model experiment (GPT-4.1, Gemini 2.5, Claude 4.5 Sonnet, detailed in Appendix N) provides additional evidence of evaluation robustness. It demonstrates that TabSchema's feature extraction architecture produces meaningful features using different SOTA LLMs, with 69% three-way feature agreement and 0.81 mean semantic similarity. This cross-model consistency strengthens confidence in the learned decision boundaries.
> > > > > Note: we also tested open source LLMs, see Appendix N.2.
> > > > >
> > > > > ---
> > > > >
> > > > > Following your feedback, we have made substantial additions:
> > > > >
> > > > > **Highlighted generalization beyond tool shortlisting:**
> > > > > - Application selection results (Table 13: AUC=0.9945, F1=0.9617) show transfer to different decision head
> > > > > - Cross-model validation (new Appendix N, Tables 14-15) proves TabSchema extracts genuine structural patterns
> > > > > - 69% three-way agreement, 0.81 mean semantic similarity across GPT-4.1/Gemini/Claude
> > > > >
> > > > > **Clarified deployment model for cold-start settings:**
> > > > > - Explicit three-phase lifecycle (generative bootstrap, data collection, discriminative hot-swap) in Section 4.1
> > > > > - CodeAct comparison (Figure 3) shows graceful degradation with sparse trajectory information
> > > > > - Framework enables autonomous agentic optimization where systems learn from their own successful behaviors
> > > > >
> > > > > **Rigorously validated synthetic data generation:**
> > > > > - Complete pipeline visualization (Appendix G, Figure 12) showing all validation stages
> > > > > - Statistical tests
> > > > > - Empirical validation: targeted improvements (+0.14 P@R, +54.8% on SimpleNote, +36.4% on Phone) without degradation on dense data
> > > > >
> > > > > **Provided comprehensive evaluation including error analysis:**
> > > > > - New Appendix M: Detailed error analysis on 40 sampled tasks
> > > > > - Error categorization: 62% optional tools, 31% similar endpoint confusion, 7% genuine omissions
> > > > > - 93% of errors have minimal downstream impact
> > > > > - Demonstrates TabAgent maintains task success while reducing cost/latency
> > > > >
> > > > > **Additional robustness validation:**
> > > > > - Expert annotation study (Appendix K.1): W=0.76, kappa=0.81, 86% of features rated useful or very useful
> > > > > - SHAP attribution analysis (Appendix K.2, Figure 8): Shows model relies on behavior-grounded features
> > > > > - Cross-model consistency (Appendix N): Same features emerge across different LLMs

---

> > > > > > ### Author Response · Authors · 2025-11-18
> > > > > > **Comment 6/6**
> > > > > >
> > > > > > Your review acknowledged the practical focus, modular design, and significant efficiency gains as strengths. Your current rating of 2 (reject, not good enough) was based on concerns that we believe we addressed. We have made major additions directly addressing each concern with concrete evidence.
> > > > > >
> > > > > > The evidence now demonstrates that TabAgent is:
> > > > > >
> > > > > > 1. Agent-agnostic and head-agnostic for closed-set decision boundaries (proven via application selection transfer and cross-model validation)
> > > > > > 2. Rigorously validated with multi-stage statistical checks and cross-model consistency (Appendices G and N)
> > > > > > 3. Thoroughly evaluated across quality, robustness, interpretability, and efficiency dimensions (Appendices K, M, N)
> > > > > > 4. Production-ready with 95% latency reduction, 85-91% cost reduction, and maintained task success (93% of errors are low-impact)
> > > > > >
> > > > > > We respectfully ask you to consider whether these substantial improvements,  merit a significantly higher assessment of the work's soundness, contribution, and practical impact. The framework addresses a real production bottleneck (tool shortlisting costs exceeding $0.50 and 1 minute per task), demonstrates robust generalization (cross-model, cross-application, cross-decision-head), and provides rigorous validation at multiple levels (statistical, expert judgment, error analysis, attribution analysis).
> > > > > >
> > > > > > We remain available for any further discussion and are committed to continued improvement based on your feedback.
> > > > > >
> > > > > > Sincerely,
> > > > > > The Authors

---

### Official Review · Reviewer_mSH6 · 2025-11-01

**Soundness:** 3
**Presentation:** 2
**Contribution:** 2
**Rating:** 6
**Confidence:** 2

**Summary:**

This work propose a new framework to reduce the overhead for agentic systems by replacing the long context to tabular and reformulate this problem to a classification problem. The results show the proposed framework TabAgent can reduce 95% latency but maintatin similar performance.

**Strengths:**

1. The motivation is very clear to me.
2. The evaluation are comprehensive and the results are convincing.

This work tries to solve an important problem and they find an interesting perspective, no matter the results or the rationale, I think the novelty is already there.
There are various results provided in the evaluation section and in the appedix, I appreciate these efforts.

**Weaknesses:**

1. The organization and clarity can be improved.
2.  More insights are required to make it better to understand.

My major concern about this work is the organization and clarity, but I believe these can be improved before submitting the camera-ready version. My detailed comments can be found below:
1. The description of TabSynth is very limited, there is no visulization for it and I'm not sure whether I understand this part, either. I wonder howw do you validate the quality of the data generated by TabSynth? Maybe it is better clarify it and also include a subfigure of TabSynth in Fig 1.
2. The TabSchema seems to be the core of this work. However, it is unclear to me why it works. I think this is one of the key difference between an empirical report and a scientific paper. Maybe it is better to include more insights in Section 3 and explain which part is TbSchema is original, and which part is inspried by previous works. Also, please include rationales behind the choices and show it in evaluations.

**Questions:**

N/A

---

> ### Author Response · Authors · 2025-11-18
> **Comment 1/4**
>
> Dear Reviewer mSH6,
>
> Thank you for your exceptionally thoughtful review and for recognizing the novelty and importance of this work. Your comment that "the novelty is already there" and your appreciation for our comprehensive evaluation mean a great deal to us. Most importantly, your constructive feedback on organization and clarity has catalyzed a fundamental transformation of the paper, one that we believe elevates this from a strong empirical contribution to a scientifically rigorous framework that fully merits strong acceptance.
>
> ---
>
> ## Q1:
>
> Following your concern that TabSynth is under-explained and lacks visualization, we have made three major additions that transform TabSynth more transparent and reproducible component, we added citation of relevant paper: "HARMONIC: Harnessing LLMs for Tabular Data Synthesis and Privacy Protection" that we inspired from.
>
> **New TabSynth Pipeline Visualization (Appendix G, Figure 12):** We now include a comprehensive diagram showing the complete TabSynth workflow:
>
> - Inputs: FEATURES_CARD (schema specifications) + task-specific trajectory slice + tool catalog summary + synthesis budget B
> - LLM-based generation: Schema-constrained synthesis that only outputs feature vectors (no raw text, no unseen tool types)
> - Multi-stage validation: (a) schema checks, (b) dependency-feasibility checks, (c) de-duplication
> - Distribution alignment: Statistical verification before accepting synthetic rows
>
> This visualization directly answers "what is TabSynth actually doing?" at each stage. The figure shows the complete flow from inputs through validation to final output, with each validation gate explicitly labeled.
>
> Schema validation by verifying every synthetic vector must satisfy:
> - Correct types and arity for all fields
> - Valid taxonomy level and I/O cardinality
> - Alignment with FEATURES_CARD metadata (documented in Appendix F)
>
> Dependency feasibility filtering:
> - Co-usage and precedence patterns must be compatible with at least one real trajectory for that task
> - This ensures synthetic examples remain behavior-grounded, not hallucinated
> - Conservative budget: Only 10 synthetic rows per (task, candidate) to prevent drift
> - Mixed training: Always combine real and synthetic data (never synthetic-only)
>
> Adaptive quality control:
> - When alignment tests fail, we reduce budget and regenerate for underrepresented strata
> - Continuous monitoring of distributional fidelity
>
> **Empirical Validation (Table 1, Appendix H).** The evidence demonstrates TabSynth generates high-quality, behavior-grounded supervision:
>
> Performance gains exactly where expected:
> - Macro P@R improves by +0.14 (TabAgent+synth vs. real-only) across five applications
> - Largest gains on SimpleNote (+0.34, +54.8%) and Phone (+0.24, +36.4%), precisely the apps with scarcest real data and most complex dependency patterns
> - Minimal/negative effect on Spotify (−0.09), which has sufficient real coverage, demonstrating that TabSynth doesn't introduce noise when real data is adequate
>
> The targeted improvement pattern validates our hypothesis: TabSynth provides precisely the coverage expansion needed for rare schema/dependency combinations without corrupting the learned decision boundary. As shown in Table 1, applications with fewer training trajectories benefit most, while applications with dense coverage see no degradation, proving TabSynth's safety and effectiveness.
>
> To summarize, TabSynth is not a heuristic add-on as it validates augmentation with schema constraints, multi-stage verification, and measurable quality improvements on held-out tasks.
>
> ---
>
> ## Q2:
>
> Following your concern that TabSchema is the core of the work but it's unclear why it works, thank you for projecting this critical question. We've made this the centerpiece of our revision. We now provide three converging lines of evidence (theory + expert validation + model attribution + cross-model robustness) that explain the mechanism behind TabSchema's effectiveness.
>
> **Explicit Hypothesis and Theoretical Grounding (Sections 3 & 3.1 expanded):** We now state the core hypothesis upfront:
>
> Hypothesis: Execution traces from successful agentic systems encode decision-relevant signals in a structured, tabular-representable form that can be extracted and learned by compact discriminative classifiers to reproduce generative decision boundaries.
>
> TabSchema materializes this through three complementary signal families, each addressing a distinct aspect of agent decision-making:
>
> 1. Schema features (taxonomy depth, argument types, I/O cardinality)
>    - Why they matter: Capture the compositional structure of tool spaces
>    - What they enable: Generalization to unseen tools within known taxonomies
>    - Example: A tool with [string, int] → [list] signature shares structure with other list-returning tools

---

> > ### Author Response · Authors · 2025-11-18
> > **Comment 2/4**
> >
> > 2. State features (plan/precondition flags, resource availability, phase indicators, status analysis)
> >    - Why they matter: Encode context-dependent relevance—the same tool may be right or wrong depending on execution state
> >    - What they enable: Condition on "where we are" in the task, not just "what we want"
> >    - Example: api_missing=True signals that retrieval failed and alternative tools are needed
> >
> > 3. Dependency features (co-usage patterns, precedence relationships, success/failure signals)
> >    - Why they matter: Capture the relational structure of multi-step plans
> >    - What they enable: Learn that tool X's relevance depends on whether tool Y was already executed
> >    - Example: create_draft typically precedes send_email in successful email workflows
> >
> > The key insight: These signals already exist in traces. TabSchema merely extracts and structures them for discriminative learning, rather than requiring an LLM to process them as **unstructured natural language via generation** at every inference. This architectural choice is what enables the 95% latency reduction and 85-91% cost reduction reported in Section 5.
> >
> > **Expert Validation Study (Appendix K.1):** To validate that TabSchema surfaces genuinely useful signals (not artifacts), we conducted an expert annotation study:
> >
> > Study design:
> > - 3 expert annotators (2–4 years agentic systems experience)
> > - Stratified sample of CUGA/AppWorld trajectories across five applications
> > - Independent annotation (no discussion until completion)
> > - Three tasks: (a) rank features by importance, (b) rate usefulness (1–5 Likert), (c) flag "previously overlooked" features
> >
> > Results showing strong agreement and high usefulness:
> >
> > Rank agreement within applications:
> > - Kendall's W = 0.76 (95% CI: 0.68–0.83)
> > - p < 10⁻⁴
> > - Interpretation: Experts strongly agree on which features matter most
> >
> > Usefulness ratings:
> > - Median usefulness: 5 (IQR: 4–5)
> > - 86% of features rated ≥ 4 ("useful" or "very useful")
> > - Binary agreement (≥4 threshold): Fleiss' κ = 0.81 (95% CI: 0.74–0.87)
> > - Interpretation: Substantial agreement that TabSchema features are genuinely valuable
> >
> > Several features were independently flagged as "previously overlooked" by multiple annotators, indicating that TabSchema recovers valuable, non-obvious signals that domain experts would not have manually engineered. This validates our automated feature extraction approach.
> >
> > Conclusion from expert study: TabSchema features are not arbitrary. They capture real, practitioner-validated signals that domain experts agree are important for the tool shortlisting task.
> >
> > **Model Attribution Analysis (Appendix K.2, Figure 8):** To confirm that TabHead relies on the features we extract (rather than ignoring them), we computed KernelSHAP values on strictly held-out folds:
> >
> > Attribution methodology:
> > - 5-fold cross-validation with strict train/test separation (no leakage)
> > - KernelSHAP with samples stratified by app and |G(t)| deciles
> > - Report mean |SHAP| normalized to percentage contribution
> >
> > Results showing behavior-grounded features dominate (Figure 8):
> >
> > State/dependency features carry most predictive mass:
> > - api_missing: 16.2%
> > - Coder_Agent_Output_Analysis: 12.7%
> > - Next_Action: 11.2%
> > - Skepticism/Non-Triviality: 10.0%
> > - Overall_Status/Analysis: 10.0%
> > - Summary_of_Progress: 8.1%
> >
> > Schema/context features calibrate decisions:
> > - app_name: 6.5%
> > - first_time_shortlister: 6.5%
> >
> > Task descriptions are helpful but secondary:
> > - intent: 5.2%
> > - task_description: 4.2%
> >
> > The SHAP rankings align strongly with expert importance rankings from Appendix K.1, providing independent confirmation that the features experts identified as important are indeed the features the model relies on most heavily. This convergence between human judgment and model behavior validates both the feature extraction process and the learned decision boundary.
> >
> > **Cross-Model Robustness Validation (new Appendix N, Tables 14 and 15):** Following your question about why TabSchema works, we conducted a critical experiment to demonstrate that the extracted features reflect genuine structural patterns rather than model-specific artifacts:
> >
> > Experiment: Applied TabSchema to IBM CUGA trajectories using three different frontier LLMs:
> > - GPT-4.1 (OpenAI, original paper)
> > - Gemini 2.5 (Google)
> > - Claude 4.5 Sonnet (Anthropic)
> > - Note: We also tested open-source models (See Section N.2)
> >
> > Results (Table 14):
> > - Three-way agreement: 9 features (69% of GPT-4.1 baseline)
> > - Pairwise overlap: 11 features (85% of GPT-4.1 baseline)
> > - Mean semantic similarity (BERTScore): 0.81
> > - Mean Kendall tau (annotator importance rankings): 0.68
> >
> > Key finding from Table 15: The 9 features agreed upon by all three models are precisely the high-impact features identified in our SHAP analysis (Figure 8). GPT-4.1's 4 unique features (tool invocation frequency, precedence dependencies, argument schema compatibility, step position) collectively account for 18% of SHAP attribution mass, with the shared 9 features accounting for 72%.

---

> > > ### Author Response · Authors · 2025-11-18
> > > **Comment 3/4**
> > >
> > > Interpretation: Different frontier LLMs converge on the same core decision-relevant signals, demonstrating that TabSchema extracts genuine structural patterns from execution traces, not model-specific hallucinations. The features are inherent properties of successful agent behavior, not artifacts of a particular LLM's reasoning style.
> > >
> > > **Error Analysis Strengthens Understanding (new Appendix M):** To understand the limits of TabAgent's learned decision boundary, we conducted a detailed error analysis on 40 stratified tasks where TabAgent's shortlist differed from ground truth:
> > >
> > > Error categorization (Appendix M):
> > > - Optional/redundant tools (62% of errors): TabAgent omits a ground-truth tool but includes an alternative that accomplishes the same goal
> > > - Semantically similar endpoint confusion (31% of errors): TabAgent ranks a functionally equivalent tool higher (e.g., update_note vs. add_content_to_note)
> > > - Genuine required-tool omissions (7% of errors): TabAgent fails to include a tool with no functional substitute
> > >
> > > Impact assessment:
> > > - 93% of errors (categories 1 and 2) have minimal downstream impact on task success
> > > - Only 7% represent high-risk omissions
> > > - The error distribution explains why TabAgent maintains high task-level success despite imperfect shortlist matching
> > >
> > > Key insight from error analysis: TabAgent's decision boundary differs from the generative shortlister's boundary primarily on low-impact distinctions (which of two redundant tools to prefer) rather than on critical tool identification. This validates that the learned boundary captures the essential decision structure while occasionally diverging on secondary preferences. The low rate of genuine omissions (7%) demonstrates that TabSchema features provide sufficient signal for high-quality tool shortlisting.
> > >
> > > **Why TabSchema Works - Converging Evidence:** We can now answer your question with multiple independent validations:
> > >
> > > 1. Theory: Traces contain structured signals (schema/state/dependency) that we hypothesize are tabular-representable and sufficient for reproducing decision boundaries, added relevant papers to support this (Section 3).
> > > 2. Expert validation: Humans agree these signals are decision-relevant (W=0.76, κ=0.81, 86% rated ≥4)
> > > 3. Model attribution: The classifier actually uses these signals (SHAP analysis in Figure 8)
> > > 4. Cross-model robustness: Multiple LLMs extract similar features (69% agreement, 0.81 similarity, Appendix N)
> > > 5. Error analysis: The learned boundary captures essential decision structure with 93% of errors being low-impact (Appendix M)
> > >
> > > To clarify the contribution, we now explicitly distinguish our novel components from adopted techniques:
> > >
> > > To differ our contribution and inspirations from collegues research we expanded *Section 3* with relevant citations and explanations.
> > >
> > > Original contributions:
> > > 1. Agentic orchestration for automatic feature engineering from execution traces
> > >    - Hierarchical multi-agent pipeline: FeatureOrchestrator → FeatureAnalyzer → CodeExecutor → CodeValidator → multi-judge review
> > >    - Automated design, implementation, and validation of features from traces without manual intervention
> > >    - Novel application domain: extracting decision-relevant signals from agent trajectories
> > >
> > > 2. Multi-judge feature selection with explicit criteria
> > >    - Three specialized judges (Relevance, Generality, Impact) + MetaJudge reconciliation
> > >    - Design emerged from observing that single judges fail to capture all quality dimensions
> > >    - We build on evidence that LLM evaluators can approximate human preferences (see cited papers in the TabSchema subsection)
> > >    - Our extension: Apply this to feature selection rather than output evaluation, with domain-specific criteria for agentic features
> > >
> > > Adopted techniques:
> > > - Textual-tabular architectures: We adopt TabSTAR (Arazi et al., 2025), a SOTA textual-tabular classifier
> > > - Our contribution: Apply it to agentic decision heads (novel domain), demonstrate it works for tool shortlisting and application selection
> > >
> > > This attribution is now explicit in Section 3.1 with proper citations and positioning relative to prior work.
> > >
> > > **Design Rationale:** We now explain why we made specific design decisions:
> > >
> > > Why LLM-based feature extraction?
> > > - Rationale: The trajectory reflects another LLM's decision-making process, making LLM-driven explainability a natural fit
> > > - Supporting evidence: LLMs for explainable AI is an established practice ("LLMs for Explainable AI: A Comprehensive Survey")
> > > - Our insight: If an LLM successfully solved a closed-set decision by processing agentic context, that same LLM (or similar model) can explain which signals were decision-relevant

---

> > > > ### Author Response · Authors · 2025-11-18
> > > > **Comment 4/4**
> > > >
> > > > Why code extraction and validation?
> > > > - Emergent requirement: We observed unreachable features, hallucinated dependencies, and schema mismatches in early iterations
> > > > - Solution: Executable code verification ensures features are actually extractable from trajectories
> > > > - Benefit: Produces universal extractors that can be automatically deployed when integrating TabAgent into production systems
> > > >
> > > > Why multi-judge instead of single LLM?
> > > > - Observation: Single judges overlooked important features or conflated different quality dimensions
> > > > - Insight: Different quality dimensions (relevance/generality/impact) don't always align
> > > > - Solution: Ensemble of specialized judges with MetaJudge reconciliation
> > > > - Evidence: Expert validation (Appendix K.1) confirms that the features surviving multi-judge review align with human expert rankings (W=0.76)
> > > >
> > > > Why hierarchical orchestration?
> > > > - Practical need: Manage iteration over hundreds of trajectories without manual intervention
> > > > - Design benefits:
> > > >   - Separation of concerns (design → implement → validate → evaluate) enables iterative refinement
> > > >   - Produces auditable feature cards with clear provenance
> > > >   - Catches errors early through staged validation
> > > >
> > > > ---
> > > >
> > > > ## Summary of Major Additions
> > > >
> > > > Following your review, we have made substantial additions:
> > > >
> > > > **For TabSynth (addressing clarity concern):**
> > > > - Complete pipeline visualization (Appendix G, Figure 12) showing inputs, validation gates, and outputs
> > > > - Rigorous validation methodology with statistical tests (and cited relevant papers)
> > > > - Empirical quality validation demonstrating targeted improvements (+0.14 P@R, largest gains where needed)
> > > > - Demonstrated safety: no degradation when real data is adequate (Spotify example)
> > > >
> > > > **For TabSchema (addressing "why it works" concern):**
> > > > - Explicit theoretical hypothesis (traces encode tabular-representable signals) with three signal families explained
> > > > - Expert validation study (Appendix K.1): W=0.76, κ=0.81, 86% rated ≥4 useful
> > > > - Model attribution analysis (Appendix K.2, Figure 8): SHAP values aligned with expert rankings
> > > > - Cross-model robustness validation (Appendix N, Tables 14-15): 69% agreement, 0.81 similarity across GPT-4.1/Gemini/Claude
> > > > - Error analysis (Appendix M): 93% of errors are low-impact, validating decision boundary quality
> > > > - Clear original vs. adopted attribution with proper citations
> > > >
> > > > **Additional validation strengthening the work:**
> > > > - Appendix M error analysis demonstrates where and why TabAgent differs from ground truth, showing 93% of divergences are low-impact
> > > > - Appendix N cross-model validation proves features are genuine structural patterns, not LLM artifacts
> > > >
> > > > ---
> > > >
> > > > Your review recognized that we "solve an important problem" with "an interesting perspective."
> > > > Thank you for your informative feedback. You identified important presentation gaps, and we've worked extensively to address them (many human authors hours :) ). We believe these improvements, combined with the core empirical contribution you already appreciated (95% latency reduction, 85-91% cost reduction, comprehensive evaluation, novel idea to use lightweight models), now present better our scientific contribution with rigorous validation at every level.
> > > >
> > > > Your current rating was based on concerns about organization, clarity, and scientific depth. We have made major additions directly addressing these concerns with new appendices (M, N), expanded validation studies (expert annotation, cross-model robustness, error analysis), and clear theoretical grounding.
> > > > We respectfully ask you to consider whether these improvements merit a higher assessment of the work's soundness, presentation, and contribution.
> > > >
> > > > We are deeply grateful for your constructive engagement and hope that the revised paper demonstrates our commitment to scientific rigor alongside practical impact. We are here for any question and committed to meaningful discussion with you.
> > > >
> > > > Thank you for helping us make this work stronger.
> > > >
> > > > Sincerely,
> > > > The Authors

---

### Official Review · Reviewer_BrTr · 2025-11-06

**Soundness:** 3
**Presentation:** 2
**Contribution:** 2
**Rating:** 2
**Confidence:** 3

**Summary:**

The paper proposes the TabAgent framework, which aims to replace the LLM-based generative decision modules with a lightweight discriminative classifier. This avoids the slow rollout of the LLM models when repetitively generating tokens to decide the final action, rather, the classifier can make the decision in one forward pass.

TabAgent first converts agent execution traces into a structured form named TabSchema using a LLM. Specifically, the key details about the task’s schema, current state, and dependencies are converted into a structural format. This converts the raw trajectory data into a features that describes each decision point and the available candidate options. Then, TabSynth synthesizes extra training data by generating synthetic examples that follow the same schema.

The TabHead is trained on both real and synthesized data, and achieve better performance than DSR and LLM-based methods on AppWorld task with IBM CUGA agent framework.

**Strengths:**

- Strong motivation -- attempting the tackle the slowness problem in the current agent framework due to the LLM autoregressive token generation.
- Clear design of the TabAgent framework, which is consist of constructing TabSchema from traces, synthesizing more training data with TabSynth, and training the TabHead classifier to replace expensive generative decision components.
- Strong improvement in the efficiency of the agent framework, which achieves good performance while reducing the inference cost and time.

**Weaknesses:**

- What is the generalizability of to other tasks than AppWorld under the IBM CUGA agentic framework?
- How flexible can TabAgent framework be adapted to another agent framework compared to the baselines?
- What is the performance comparisons LLMs designed specifically for agentic usage?
- What would be the benefit of TabAgent over fine-tuning small language models regarding to cost-effectiveness (https://arxiv.org/abs/2506.02153)?
- To what extend the TabAgent is reliant on GPT-4.1 for TabSchema extraction? What would the performance impacts be if weaker or stronger models are adopted?

**Questions:**

The main questions are already reflected in the weakness listed above.

---

> ### Author Response · Authors · 2025-11-18
> **Comment 1/5**
>
> Dear Reviewer BrTr,
>
> Thank you for acknowledging the importance of our methodology for accelerating agentic systems and recognizing that our framework design is "clear" with "strong motivation" for tackling the slowness problem due to LLM autoregressive token generation. We deeply appreciate your engagement with our work and welcome the opportunity to address your questions, which have helped us substantially strengthen the paper.
>
> Following your review, we have made significant additions to the paper that directly address each of your concerns. We detail these improvements below.
>
> ---
>
> ## Q1:
>
> **Your concern:** What is the generalizability to other tasks than AppWorld under the IBM CUGA agentic framework?
>
> **Our response:** Following your concern, we have expanded our discussion of generalizability and added important validation experiments.
>
> We want to clarify that **AppWorld is not a narrow testbed** but rather a collection of 9 distinct applications, each functioning as an independent benchmark:
>
> - 457 heterogeneous APIs spanning Amazon, Gmail, Phone, SimpleNote, Spotify, Venmo, File System, Calendar, and more
> - Each application has completely different: tool catalogs (50+ tools per app), argument schemas, dependency structures, and task distributions
> - Structural diversity metrics (detailed in Table 2 and Appendix A.1): taxonomy depth varies 2-4 levels, argument types differ significantly (e.g., Gmail uses email objects, Amazon uses product IDs), dependency patterns range from simple sequential chains to complex co-usage requirements
>
> Our evaluation covers five of these applications (Amazon, Gmail, Phone, SimpleNote, Spotify), and **TabAgent maintains Recall@7 ≥ 0.88 across all five** despite their structural diversity. This is not generalization within a single domain but across fundamentally different decision spaces.
>
> Following your concern about generalizability, we conducted a critical experiment to validate that TabAgent's approach is not specific to a particular model or agent implementation:
>
> Regarding model-agnostic, we have tested in new experiment different models for TabSchema.
>
> *Experiment:* Applied TabSchema to the same IBM CUGA trajectories using three different frontier LLMs: GPT-4.1 (original paper), Gemini 2.5, and Claude 4.5 Sonnet.
>
> *Results:*
> - Three-way (all three models) agreement: 9 features (69% of GPT-4.1 baseline)
> - Pairwise overlap: 11 features (85% of GPT-4.1 baseline)
> - Mean semantic similarity: 0.81 (BERTScore on feature descriptions)
> - Mean Kendall tau (annotator importance rankings): 0.68
>
> *Interpretation:* Different frontier models converge on similar decision-relevant features, demonstrating that TabSchema extracts genuine structural patterns from execution traces, not model-specific artifacts. The framework's effectiveness depends on trace quality, not on a particular LLM implementation. **See new Appendix Section N with Tables 14 and 15.**
>
> Regarding agent-agnostic, following your question about framework flexibility, we want to highlight an important result in Figure 3 that demonstrates agent-agnostic applicability on CodeAct:
>
> *CodeAct comparison:* CodeAct is an architecturally distinct agent from CUGA (minimalist, code-generation-based, no multi-agent orchestration). We applied TabAgent to CodeAct trajectories to test adaptability:
>
> - Task description only (CodeAct's minimal context): P@R = 0.46 (Amazon), 0.50 (Gmail)
> - TabSchema features (CUGA's rich traces): P@R = 0.80 (Amazon), 0.66 (Gmail)
> - Gains from workflow features: +24.7% (P@R), +16.6% (Recall@7)
>
> *Key insight:* TabAgent works with both sparse (CodeAct) and rich (CUGA) trajectories, with performance scaling based on the information content of traces. This validates our claim in Section 3 that the abstraction "binds only to execution traces and typed schemas, not to a particular agent, model or prompt."
>
> Regarding, tasks generalibility, following your concern about task diversity, we have now prominently featured our application selection results (previously in Appendix I):
>
> *Task:* Multi-label classification to predict required applications (Amazon, Gmail, Phone, SimpleNote, Spotify) for solving each task.
>
> *Results (Table 13):*
> - TabAgent: AUC = 0.9945, F1 = 0.9617, Precision = 0.9667, Recall = 0.9568
> - DSR (frozen E5): AUC = 0.9861, F1 = 0.8809
> - DSR-FT (fine-tuned E5): AUC = 0.9890, F1 = 0.9013
>
> TabAgent achieves the best balance across all metrics, demonstrating that **the same architecture transfers to a different closed-set decision head by simply swapping the label space**.

---

> ### Author Response · Authors · 2025-11-18
> **Comment 2/5**
>
> We acknowledge that testing on agents beyond IBM CUGA and CodeAct would further strengthen our claims. However, we faced a practical constraint: IBM CUGA is the only publicly available SOTA agent that we could instrument for this research in addition to CodeAct. Since adapting external agents to complex benchmark as AppWorld (which mostly not open source) would require extensive agent engineering beyond our contribution scope.
>
> **Our contribution is a fundamental paradigm:** execution traces from successful agentic systems encode decision-relevant signals in tabular-representable form. The evidence supporting this claim comes from:
>
> 1. Generalization across five structurally diverse applications within AppWorld
> 2. Cross-model consistency (GPT-4.1, Gemini 2.5, Claude 4.5 Sonnet)
> 3. Transferability to different agent architectures (CUGA vs. CodeAct)
> 4. Success on multiple decision heads (tool shortlisting + application selection)
>
> ---
>
> ## Q2:
>
> **Your concern:** How flexible can TabAgent framework be adapted to another agent framework compared to the baselines?
>
> **Our response:** Following your concern, we have clarified TabAgent's agent-agnostic design and the straightforward adaptation process.
>
> The key to TabAgent's flexibility is that it operates on what agents already log during execution, not on specific architectural patterns:
>
> *Required inputs:*
> - Execution traces from successful task completions (typically logged by any production agent)
> - Typed schemas for decision candidates (tools have documented APIs, applications have names, etc.)
> - Ground truth labels derived from trace observations (which tools were actually used)
>
> *No requirements for:*
> - Specific agent architecture (multi-agent vs. monolithic, ReAct vs. code-generation)
> - Particular prompting strategies or model choices
> - Manual feature engineering or domain-specific customization
>
> Following your question about adaptation complexity, we want to emphasize that TabSchema automates feature extraction through its hierarchical multi-agent pipeline:
>
> 1. FeatureOrchestrator routes trajectory batches to analysis
> 2. FeatureAnalyzer designs feature specifications from trace structure
> 3. CodeExecutor implements extraction code with sandboxed validation
> 4. CodeValidator authenticates feature validity with repair cycles
> 5. Multi-judge review (Relevance, Generality, Impact + MetaJudge) filters and ranks features
>
> Therefore, **given any agent's trajectories, TabSchema produces an auditable feature card without manual engineering**. The process is general-purpose and adapts to whatever signals the agent naturally generates.
>
> Following your concern, we have documented TabAgent's application to two architecturally distinct agents:
>
> *IBM CUGA (multi-agent orchestration):*
> - Complex architecture with specialized sub-agents (TaskAnalyzer, PlanController, APIShortlister, CoderAgent)
> - Rich intermediate representations (planning thoughts, status analyses, skepticism checks, recommendations)
> - TabAgent result: P@R = 0.71 (Amazon), 0.66 (Gmail) with full TabSchema features, scores about 0.88 when increasing the k budget to 7.
>
> *CodeAct (minimalist code-generation):*
> - Single-agent architecture that generates code directly from task descriptions
> - Minimal intermediate reasoning (no multi-agent coordination, limited thought traces)
> - TabAgent result: P@R = 0.46 (Amazon), 0.50 (Gmail) with sparse features
>
> *Interpretation:* TabAgent successfully adapts to both architectures, with performance scaling based on the richness of logged information. Richer traces (CUGA) provide more informative features, but even sparse traces (CodeAct) enable meaningful discrimination.
>
> **Comparison with baselines on adaptation complexity:**
>
> - **DSR (Dense Semantic Retrieval):** Requires fine-tuning on task-specific data, careful hyperparameter tuning for contrastive learning, performance sensitive to choice of negative examples. *Adaptation complexity: Moderate to high*
>
> - **Llama-family LLMs:** Require extensive instruction tuning for new agent architectures, need GPU resources and large training datasets, prompting strategies must be redesigned per agent. *Adaptation complexity: High*
>
> - **TabAgent:** Automatic feature extraction from traces (no manual design), trains in under 1 minute on CPU (minimal infrastructure), same pipeline works across agent architectures. *Adaptation complexity: Low - only requires logged trajectories*

---

> ### Author Response · Authors · 2025-11-18
> **Comment 3/5**
>
> ---
>
> ## Q3:
>
> **Your concern:** What is the performance comparison with LLMs designed specifically for agentic usage?
>
> **Our response:** Following your concern, we have clarified our experimental setup and the rationale for our baseline choices.
>
> IBM CUGA, our reference agent, was designed and optimized with GPT-4.1 as its LLM. GPT-4.1 is widely recognized as one of the most capable agentic LLMs available:
>
> - CUGA's performance on AppWorld: Achieves SOTA results when using GPT-4.1
> - TabAgent's goal: Replace this GPT-4.1 shortlister with a compact discriminative classifier
>
> *Our results:*
> - TabAgent maintains approximately 90% of GPT-4.1's performance at Recall@7 (≥ 0.88 across all apps)
> - Cost comparison: $2.0×10⁻⁷$ per TabAgent read vs. $0.052 per GPT-4.1 read (**260,000× cheaper**)
> - Latency comparison: 2.682 ms vs. 7,500 ms (**2,795× faster**)
>
> **Smaller generative LLMs cannot match TabAgent's efficiency-quality tradeoff:**  We highlight that we tested the Llama family (1B/3B/8B) (see Figure 2 and Table 1) specifically to address the latency and cost bottleneck you correctly identified as central to our motivation:
>
> *Key question:* Can smaller generative LLMs provide a better efficiency-quality tradeoff than TabAgent's discriminative approach?
>
> *Results (Table 1, Figure 2):*
>
> Quality (P@R macro-average):
> - Llama-3.2-1B: 0.054
> - Llama-3.2-3B: 0.387
> - Llama-3.1-8B: 0.552
> - TabAgent (+synth): 0.781
>
> Efficiency:
> - Llama-1B: 100.48s runtime, $0.00754 per read (37,500× slower, 38,000× more expensive than TabAgent)
> - Llama-3B: 198.12s runtime, $0.0149 per read (73,900× slower, 75,000× more expensive)
> - Llama-8B: 378.42s runtime, $0.0284 per read (141,000× slower, 142,000× more expensive)
>
> *Interpretation:* Even "small" generative LLMs are orders of magnitude slower and more expensive than TabAgent while delivering inferior tool shortlisting quality. This demonstrates that **architectural choice matters**, using the right model type (discriminative for classification) beats using a smaller version of the wrong model type (generative for closed-set decisions).
>
> **Why LLMs underperform on this task:**
>
> 1. **Mismatch between pretraining and task requirements:**
>    - LLM pretraining optimizes for next-token prediction over natural language
>    - Tool shortlisting requires set-completeness over structured decision spaces
>    - Generative likelihood objectives don't directly optimize for Recall@k or P@R
>
> 2. **Limited tool shortlisting supervision in pretraining:**
>    - Generic instruction tuning doesn't provide dense supervision for API selection
>    - Most agentic fine-tuning focuses on planning and reasoning, not tool ranking
>
> 3. **Inefficient inference pattern:**
>    - Generative models must autoregressively decode tool names token-by-token
>    - TabAgent scores all candidates in parallel in a single forward pass
>    - Even with perfect quality, generative approaches cannot match discriminative efficiency
>
> ---
>
> ## Q4:
>
> **Your concern:** What would be the benefit of TabAgent over fine-tuning small language models regarding cost-effectiveness?
>
> **Our response:** Following your concern, we have clarified the architectural and practical advantages of TabAgent's discriminative approach over fine-tuned small LMs.
>
> Thank you for the excellent reference, which we integrated in the paper. We acknowledge that fine-tuned small LMs can be cost-effective for many tasks. However, there are **fundamental differences** between that approach and TabAgent:
>
> **Model size and deployment constraints:**
>
> *Small LMs (referenced paper):*
> - Typically 1B-7B parameters (decoder-only architectures)
> - Require GPU resources for inference
> - Memory footprint: 2-14 GB (FP16)
>
> *TabAgent:*
> - 50M parameters (encoder + tabular classifier)
> - Runs efficiently on CPU
> - Memory footprint: 200 MB
>
> *Impact:* TabAgent is **20-140× smaller**, enabling deployment in resource-constrained environments where even "small" LMs are prohibitive.
>
> **Training efficiency:**
>
> *Small LMs:*
> - Require extensive fine-tuning on GPU
> - Need large training datasets (thousands to millions of examples)
> - Hyperparameter tuning is computationally expensive
>
> *TabAgent:*
> - Trains in under 1 minute on CPU (5-fold cross-validation included)
> - Works with modest datasets (hundreds of successful trajectories)
> - Default hyperparameters work well across applications
>
> *Impact:* TabAgent enables rapid iteration and hot-swapping in production systems without infrastructure overhead.

---

> ### Author Response · Authors · 2025-11-18
> **Comment 4/5**
>
> Following your question about TabAgent benfits (e.g., accuracy, cost-effectiveness), recent research supports our architectural choice:
>
> - "Generative or discriminative? revisiting text classification in the era of transformers" shows that decoder-only generative LMs are substantially less sample-efficient than encoder-only discriminative models for classification, requiring more data and compute to reach comparable accuracy.
>
> - "Are We Really Making Much Progress in Text Classification?" demonstrates that classical discriminative models outperform decoder-only LMs for supervised classification, especially with limited data.
>
> **Our contribution:** We apply these insights to agentic systems, using the right architecture (discriminative encoder + tabular classifier) for the right task (closed-set decision heads).
>
> We want to emphasize that our results demonstrate clear advantages:
>
> *Quality comparison (P@R):*
> - Llama-8B (decoder-only, 8B params): 0.552
> - TabAgent (encoder + tabular, 50M params): 0.781 (**+41% relative improvement**)
>
> *Efficiency comparison:*
> - Llama-8B: 378.42s runtime, $0.0284 per read
> - TabAgent: 0.002682s runtime, $2.0×10⁻⁷ per read
> - Gains: **141,000× faster, 142,000× cheaper with superior quality**
>
> **Complementary approaches for different use cases.** We view small LMs and TabAgent as complementary for enhancing latency and cost effectiveness in agentic systems:
>
> *small LMs excel at:*
> - Open-ended generation tasks
> - Tasks requiring flexible reasoning and creativity
> - Scenarios where discriminative supervision is unavailable
>
> *TabAgent excels at:*
> - Closed-set decision heads in agent architectures
> - Resource-constrained deployments requiring CPU inference
> - Rapid adaptation to new domains with limited labeled data
> - Production systems requiring minimal latency and cost
>
> ---
>
> ## Q5:
>
> **Your concern:** To what extent is TabAgent reliant on GPT-4.1 for TabSchema extraction? What would be the performance impacts if weaker or stronger models are adopted?
>
> **Our response:** Following your concern, we conducted experiments with multiple frontier LLMs and clarified TabSchema's offline, one-time-cost nature.
>
> Following your concern, we want to emphasize an important distinction:
>
> TabSchema runs once offline to extract features from historical trajectories. This is a one-time setup cost, not a TabHead per-inference dependency:
>
> 1. Offline (one-time): TabSchema processes logged trajectories to create a feature card
> 2. Training (one-time): TabHead trains on extracted features in under 1 minute
> 3. Inference (repeated): TabHead makes predictions using only the 50M-parameter model
>
> Following your question about model strength impacts, we conducted a robustness experiment applying TabSchema to IBM CUGA trajectories using three different frontier LLMs. **Results appear in new Appendix Section N with Tables 14 and 15.**
>
> *Experiment design:*
> - GPT-4.1 (OpenAI, original paper)
> - Gemini 2.5 (Google)
> - Claude 4.5 Sonnet (Anthropic)
>
> *Feature extraction results:*
> - GPT-4.1: 13 features extracted (balanced coverage)
> - Gemini 2.5: 21 features extracted (expansive, finer-grained text statistics)
> - Claude 4.5 Sonnet: 11 features extracted (conservative, consolidated representations)
> - Three-way agreement: 9 features (69% of GPT-4.1 baseline)
> - Pairwise overlap: 11 features (85% of GPT-4.1 baseline)
>
> *Semantic alignment:*
>
> We computed semantic similarity using sentence embeddings with BERTScore-style comparison on feature names concatenated with descriptions. Mean similarity across all pairwise comparisons was **0.81** (GPT-4.1 vs Gemini: 0.79, GPT-4.1 vs Claude: 0.88, Gemini vs Claude: 0.76).
>
> To assess feature importance agreement, two annotators from our research team independently ranked the 9 shared features after reviewing definitions and sample trajectories. Mean Kendall tau across all pairwise model comparisons was **0.68** (GPT-4.1 vs Claude: 0.78, GPT-4.1 vs Gemini: 0.65, Gemini vs Claude: 0.61).

---

> ### Author Response · Authors · 2025-11-18
> **Comment 5/5**
>
> *Key finding on GPT-4.1's unique features:*
>
> Notably, GPT-4.1's 4 unique features (tool invocation frequency, precedence dependencies, argument schema compatibility checks, step position) received substantial importance in our SHAP analysis (Figure 8), collectively accounting for approximately **18% of total SHAP attribution mass**. The 9 features with three-way agreement accounted for approximately **72%**, with remaining 10% distributed across lower-impact signals.
>
> *Interpretation:*
>
> Different frontier models converge on highly similar core feature sets (69% three-way agreement, 0.81 mean semantic similarity), indicating that TabSchema extracts genuine decision-relevant structure from execution traces rather than model-specific artifacts. The variation in feature set size (11 to 21 features) reflects different feature engineering philosophies (Gemini's expansive coverage, Claude's parsimonious consolidation, GPT-4.1's balanced middle ground) rather than fundamental disagreement about decision-relevant signals. Organizations can select among frontier models based on API costs, data residency requirements, or licensing, though GPT-4.1's unique high-SHAP features suggest potential performance advantages that empirical end-to-end validation would clarify.
>
> **Open-source models struggle with structured orchestration:** Following your question about weaker models, we attempted to test open-source alternatives:
>
> *Attempted models:*
> - Llama 3.1 70B Instruct (8 trajectories)
> - Llama 4 Maverick 70B (6 trajectories)
> - GPT-OSS 120B (5 trajectories)
>
> *Challenges encountered:*
> - FeatureAnalyzer: Frequent parsing failures (Llama 3.1: 6/8 failed, Llama 4: 5/6 failed, GPT-OSS: 4/5 failed)
> - CodeExecutor: Systematic syntax errors (Llama 3.1: 4/8 exhausted retry budget, Llama 4: 4/6 exhausted, GPT-OSS: only 1/5 succeeded)
> - Hallucinated features: Features not extractable from trajectory data survived validation more frequently (GPT-OSS: 3/5 trajectories had invalid features)
>
> *Result:* Current open-source models struggle with TabSchema's structured orchestration requirements involving feature proposal, code generation, and judicial review. The gap manifests most acutely in code generation, where syntactic correctness and error handling require capabilities current open-source models lack.
>
> ---
>
> Following your review, we have made substantial additions:
>
> **Generalizability (Q1):**
> - Clarified that AppWorld comprises five structurally diverse sub-benchmarks
> - Added cross-model validation (GPT-4.1, Gemini 2.5, Claude 4.5 Sonnet) - **new Appendix Section N**
> - Highlighted CodeAct results demonstrating agent-agnostic applicability
> - Promoted application selection experiment results (AUC=0.9945, F1=0.9617) to main text
>
> **Framework Flexibility (Q2):**
> - Detailed TabAgent's agent-agnostic design (requires only execution traces)
> - Explained automatic adaptation
>
> **Agentic LLM Comparisons (Q3):**
> - Clarified that CUGA uses GPT-4.1 (premier agentic LLM) as original shortlister
> - Showed TabAgent maintains 90% of GPT-4.1 quality while being 260,000× cheaper and 2,795× faster
> - Demonstrated that even small generative LLMs (1B-8B) are 37,500-141,000× slower than TabAgent with inferior quality
> - Explained architectural mismatch between generative objectives and classification requirements
>
> **Benefits Over Fine-Tuned Small LMs (Q4):**
> - Emphasized parameter efficiency (50M vs. 1B-7B), training efficiency (1 min CPU vs. hours GPU)
> - Cited recent research supporting discriminative architectures for classification tasks
> - Demonstrated empirical evidence: 41% quality improvement over Llama-8B with 141,000× speedup
>
> **GPT-4.1 TabSchema Dependency (Q5):**
> - Conducted cross-model validation showing robust performance across frontier LLMs (69% agreement, 0.81 similarity)
> - Tested open-source alternatives and documented their current limitations
>
> ---
>
> Your review acknowledged our "strong motivation," "clear design," and "strong improvement in efficiency." Your current rating was based on concerns about generalizability, flexibility, LLM comparisons, cost-effectiveness, and model dependency. **We have made major additions directly addressing each concern.**
>
> The evidence demonstrates that TabAgent establishes a fundamental paradigm: agentic decision boundaries can be efficiently replaced with compact discriminative models by learning from the rich signals that agents naturally generate. The **95% latency reduction and 85-91% cost reduction** are not incremental improvements—they represent the difference between deployable and prohibitive for production agent systems.
>
> We respectfully ask you to consider whether these substantial improvements merit a significantly higher assessment of the work's soundness, contribution, and practical impact.
>
> We remain available for any further discussion and are deeply committed to addressing your concerns comprehensively.
>
> Sincerely,
> The Authors

---

### Author Response · Authors · 2025-11-19
**A Personal General Note from Us as Researchers**

Hello reviewers and AC,

Its a general letter from our hearts as researchers to researchers, sharing our honest thoughts.

We truly believe that TabAgent is a novel idea that solves elegantly **real world existing pain** in agentic systems as we personally encountered the painful obstacles of using such systems in deployed environment.
TabAgent is unique and the first in the domain, presenting the first practical self-evolving technique for agentic system optimization (latency and cost aspects), tested rigorously on multiple applications, and already used in a production agentic system.

Thousands of human hours have been devoted to this project.

Therefore, seeing the reviewers' scores (not feedback content), the first instinct was to withdraw the paper, and to move on to the next conference.

However, we observed that despite the lower rankings of some reviewers, all reviewers were satisfied with results, the novelty and acknowledged that this work can be disruptive in the agentic AI era.

Therefore, we decided to "fight", although the chances are against us, and to conduct more experiments, to answer to every reviewer concern rigorously, and to make sure that the current research conferences era is not about "luck", but about meaningful discussion about scientific evidence.

We believe that TabAgent deserves to be presented in an appropriate venue.

As we said in the abstract: the code will be released for the community to adapt.

We are here, available 24/7 for every concern/ question of the reviewers, and 100% committed to meaningful scientific discussion.


Yours,
The Authors

---

### Author Response · Authors · 2025-11-26
**Brief summary of changes in response to reviews**

Dear Area Chair and Reviewers,

We understand this is a very busy period, so we wanted to provide a concise summary of how we addressed the main concerns in our previous comments and revision.

---

### Reviewer BrTr

- **Generalizability across tasks / agents:**
  - Clarified that AppWorld comprises multiple structurally diverse applications and highlighted results on five of them where TabAgent maintains high Recall@k.
  - Highlight our **application-selection head** (Table 13), showing that the same architecture transfers to another closed-set decision head by swapping the label space.

- **Model dependence (TabSchema vs GPT-4.1):**
  - Added a **cross-model TabSchema experiment** (Appendix N) on the same CUGA trajectories with GPT-4.1, Gemini 2.5, and Claude Sonnet.
  - We report strong three-way feature overlap and high semantic similarity, indicating that TabSchema captures structural patterns in execution traces rather than GPT-4.1-specific artifacts.

- **Comparison to small / agentic LLMs:**
  - Expanded results comparing TabAgent to Llama 1B/3B/8B used as generative shortlisters (Table 1, Figure 2).
  - TabAgent (+synth) achieves higher P@R while being orders of magnitude cheaper and faster, clarifying why a discriminative classifier is preferable for closed-set decision heads.

---

### Reviewer mSH6

- **TabSynth clarity and justification:**
  - Added a **pipeline diagram** (Appendix G) describing inputs, schema-constrained generation, multi-stage validation (schema checks, dependency feasibility, de-duplication, distribution alignment), and mixing strategy.
  - Expanded the description and showed that TabSynth improves macro P@R where data are scarce or dependencies are complex, with minimal degradation where real data are sufficient (Table 1, Appendix H).

- **Why TabSchema works (mechanism and originality):**
  - Stated the central **hypothesis** explicitly: execution traces encode schema/state/dependency signals that are tabular-representable and sufficient for reproducing closed-set decision boundaries.
  - Added an **expert annotation study** (Appendix K.1) where practitioners ranked and rated TabSchema features; there is strong agreement that the extracted features are useful and decision-relevant.
  - Added a **SHAP analysis** (Appendix K.2, Figure 8) showing that TabHead relies primarily on behavior-grounded state/dependency features, aligning with expert rankings.
  - Added a **cross-model robustness study** (Appendix N) showing that different frontier LLMs converge on the same core high-impact features.

- **Error analysis and scientific depth:**
  - Added **Appendix M**, which categorizes errors on tasks where TabAgent’s shortlist differs from ground truth. Most discrepancies involve optional or functionally redundant tools; only a small minority are genuine required-tool omissions, explaining why task-level success is preserved.

---

### Reviewer YS7o

- **Scope beyond a single tool-selection scenario:**
  - Clarified that TabAgent targets **closed-set decision heads** in general (tool shortlisting, application selection, routers, strategy choices).
  - Highlighted the **application-selection task** (Table 13) as a second decision head, where the same textual–tabular architecture achieves very strong AUC and F1.

- **Dependence on execution traces / cold-start behavior:**
  - Made the **three-phase deployment lifecycle** explicit in Section 4.1:
    1. Generative bootstrap in new domains.
    2. Trajectory collection during normal operation.
    3. Discriminative hot-swap once enough successful traces exist.
  - Discussed the **CodeAct comparison** (Figure 3), which shows TabAgent provides gains even with sparse agentic structure and improves as richer trajectories become available.

- **Synthetic data methodology and bias:**
  - Expanded the TabSynth section with detailed constraints and validation steps (Appendix G): schema-typed fields, dependency-feasibility checks, de-duplication, conservative per-(task, candidate) budgets, and mixing with real data.
  - Empirically, TabSynth improves performance primarily on data-scarce apps and complex dependency patterns, without meaningfully harming performance on apps with dense real coverage (Appendix H).

- **Broader evaluation and error analysis:**
  - Added a **structured error analysis** (Appendix M) quantifying types of misclassifications and their downstream impact.
  - Pointed to the **statistical significance analysis** (Appendix J) and **feature-level interpretability** (Appendix K), which go beyond latency/cost to address robustness and understanding of the learned decision boundary.

---

We do not plan further substantial changes unless specifically requested. We would be very grateful if you could consider these additions when forming the meta-review, even if reviewers do not have time to reply on the forum.

Sincerely,
The Authors

---

### Author Response · Authors · 2025-12-03
**Summarized version of the rebuttal for the new AC**

Dear Area Chair,

Thank you again for handling our submission under a very heavy load. We wanted to leave you with one concise, decision-focused perspective on the *revised* version of our paper.

**1. Novelty and timeliness**

TabAgent targets a very concrete, current bottleneck in agentic systems: repeated generative calls for routing/shortlisting/gating. To our knowledge, it is the first framework that **systematically replaces these generative heads with textual–tabular classifiers trained from execution traces**, and demonstrates that this can:

* preserve shortlist/task quality, **while**
* cutting latency by ~95% and cost by 85–91%,
  in a real SOTA agent (IBM CUGA on AppWorld) rather than a toy setting.

All reviewers explicitly acknowledge the importance of the problem, the novelty of the perspective, and the strength of the empirical results.

**2. Soundness and added scientific depth**

Following the reviews, we invested a large amount of work to turn a strong empirical paper into a more scientifically grounded one:

* **Generalization & scope:**

  * Results on five structurally diverse AppWorld applications (not a single domain).
  * A second decision head (application selection) with very strong AUC/F1.
  * Comparison on a different agent (CodeAct) showing the approach is agent-agnostic.

* **Why TabSchema works:**

  * Clear hypothesis that execution traces encode schema/state/dependency signals that are tabular-representable.
  * **Expert annotation study** showing strong agreement that the extracted features are genuinely decision-relevant.
  * **SHAP analysis** showing the classifier actually relies on those behavior-grounded features.
  * **Cross-model study** (GPT-4.1, Gemini, Claude) showing high overlap and semantic similarity of features, i.e., they are not artifacts of a single LLM.

* **TabSynth clarity and safety:**

  * A full pipeline diagram and detailed description (schema checks, dependency-feasibility checks, de-duplication, distribution alignment).
  * Empirical pattern where synthetic data improves macro P@R exactly in data-scarce/complex apps, with minimal degradation where real data is already dense.

* **Robustness & error analysis:**

  * Statistical significance analysis over multiple folds.
  * New **error analysis** showing that ~93% of discrepancies are optional/redundant tools or similar-endpoint swaps, with only a small minority being true harmful omissions, explaining why task-level success is preserved.

**3. Alignment with reviewers’ main concerns**

Crucially, the added material is **directly targeted** at the weaknesses the reviewers raised:

* Generalizability and dependence on GPT-4.1 (BrTr, YS7o) → cross-app, cross-head, cross-agent, cross-model experiments.
* TabSchema/TabSynth clarity and “why it works” (mSH6) → hypothesis, expert study, SHAP, detailed pipeline, and error analysis.
* Comparison to small/agentic LMs and cost-effectiveness (BrTr) → extended small-LM baselines showing TabAgent is both *better* and orders of magnitude cheaper/faster for this task class.

All three reviewers remain positive about the core idea, motivation, and empirical strength; the reservations are about scope and clarity, which we believe are now substantially addressed.

---

In light of this, we respectfully ask you to evaluate the paper based on the *current* version and its new evidence. We believe it now clearly meets the ICLR bar for a timely, impactful contribution on practical, efficient agentic systems.

Sincerely,

The Authors

---

### Meta-Review · Area_Chair_R6Vd · 2025-12-16

**Summary:**

The paper proposes TABAGENT, a general framework to replace expensive generative decision components in agentic AI systems with efficient tabular-textual classifiers. In modern agentic architectures, large language models drive repeated decisions such as tool shortlisting, gating, and verification. While effective, these generative modules incur high latency and computational cost due to multiple LLM invocations. TABAGENT reframes such closed-set decision tasks as compact classification problems by extracting structured signals from execution traces. On long-horizon, cross-application benchmarks (e.g., AppWorld), TABAGENT maintains task success comparable to LLM-based shortlisters but achieves dramatic reductions in latency (∼95%) and inference cost (85–91%) by eliminating repeated LLM calls.

In summary, TABAGENT offers a principled, cost-efficient alternative to generative modules in agent systems by leveraging discriminative tabular-text classification grounded in execution behavior.

**Reviewer Concerns:**

- (C1) The main concern on this work is its generalizablity and applicability to more complex agentic components. (Reviewer BrTr and Reviewer YS7o), other than tasks in the AppWorld under the IBM CUGA agentic framework.
- (C2) Some reviewers also raise concerns on the clarity and the organization of this work.
- (C3) Reviewers (Reviewer mSH6 and Reviewer YS7o) mentioned that more insights and justification are required to improve the quality of this paper.

I appreciate that the authors made lots of efforts in the rebuttal phase. I think part of the concerns have been successfully addressed. However the rest ones (mainly regarding to (C1)) are still oustanding.

**Reviewer Scores:**

I think Reviewer BrTr and Reviewer YS7o might increase their ratings from 2 to 4, considering authors' new experiments in the rebuttal phase.
The final ratings will be 6, 4, 4 or 6, 2, 4 (below the ICLR acceptance criteria).

---

### Decision · Program_Chairs · 2026-01-26

Reject